# Modeling future spread of infections via mobile geolocation data and population dynamics. An application to COVID-19 in Brazil

**Pedro S. Peixoto** *, Diego Marcondes, Cláudia Peixoto, Sérgio M. Oliva

Department of Applied Mathematics, Institute of Mathematics and Statistics, University of São Paulo, São Paulo, Brazil

* ppeixoto@usp.br

## Abstract

Mobile geolocation data is a valuable asset in the assessment of movement patterns of a population. Once a highly contagious disease takes place in a location the movement patterns aid in predicting the potential spatial spreading of the disease, hence mobile data becomes a crucial tool to epidemic models. In this work, based on millions of anonymized mobile visits data in Brazil, we investigate the most probable spreading patterns of the COVID-19 within states of Brazil. The study is intended to help public administrators in action plans and resources allocation, whilst studying how mobile geolocation data may be employed as a measure of population mobility during an epidemic. This study focuses on the states of São Paulo and Rio de Janeiro during the period of March 2020, when the disease first started to spread in these states. Meta-population models for the disease spread were simulated in order to evaluate the risk of infection of each city within the states, by ranking them according to the time the disease will take to infect each city. We observed that, although the high-risk regions are those closer to the capital cities, where the outbreak has started, there are also cities in the countryside with great risk. The mathematical framework developed in this paper is quite general and may be applied to locations around the world to evaluate the risk of infection by diseases, in special the COVID-19, when geolocation data is available.

## Introduction

The COVID-19, caused by the coronavirus SARS-CoV-2, has spread quickly after its first reported cases in Wuhan, China, in December 2019, posing a serious threat to health systems and the world economy [1]. Since March 2020, when the disease was classified by WHO as a pandemic [2], countries around the world have followed protocols implemented months before in Asia, enforcing a variety of interventions, from mild to radical ones, based on social distancing, isolation and quarantine, to slow the disease spread, as recommended by WHO [3]. It is a common sense that the pandemic should be fought in two frontiers: by saving lives while avoiding the collapse of health systems, and by protecting the population from the economic impacts of the pandemic, specially its most vulnerable parcel [2]. For either goal to be achieved, health officials and government authorities should have reliable information about

**Data Availability Statement:** All methods discussed in this work are available in python (Python 493 Software v3.6) or R (R software for statistical computing vs 3.6.3) program codes

under open-source license, available at https://github.com/pedrospeixoto/mdyn. The mobile geolocation movement matrices are also available within the same GitHub repository. Supporting information is provided at www.ime.usp.br/~pedrosp/covid-19.

**Funding:** Dr. Peixoto acknowledges the research grant of Fundação de Amparo à Pesquisa do Estado de São Paulo (FAPESP 16/18445-7) Marcondes has received financial support from Conselho Nacional de Desenvolvimento Científico e Tecnológico (CNPq) during the development of this research.

**Competing interests:** The authors have declared that no competing interests exist.

the disease spreading and its economic and social impacts, hence, for instance, the modelling of such spreading is not only a scientific achievement, but also a source of crucial strategic information. Indeed, a way of reducing the damages caused by the pandemic is to model how the disease will spread, in order to properly assign the available resources to locations where they will be needed the most.

Another strategic information governments need to have is about the efficacy of the interventions enforced to slow the disease spread. Initial reports have shown the efficacy of these interventions, but we still lack reliable data, especially in Brazil, and, even when the data is available, we need to sort out misleading information [4]. Among the several challenges to address this pandemic, detecting the spatial spread of the disease within a region is one of the top priorities. An early warning can give time for government authorities to prepare the health system in a location to endure the increase on the number of people in need of medical care. One way to overcome this challenge is to monitor human mobility in order to detect patterns from which to predict future focus of infection, to either asses the efficacy of implemented policies to avoid transmission, or drive policies with the goal of avoiding the transmission to certain locations. This monitoring, specially using mobile phone data, has been noted to be an efficient way to follow public mobility. In recent work, [5] has indicated the efficacy of the intervention in China, correlating mobile data with reported cases. In other report, mobile data has evidenced the effect of Government-enforced measures in São Paulo, Brazil, in reducing social contact [6]. It is worth noting that, for large scale movements, other measures beyond mobile phone data have been successfully used to foresee the spread of the disease in Brazil [7].

As mentioned by Brockman [8, 9], while the time evolution of the epidemics is frequently modeled in the literature by dynamical differential equations or time series [10, 11], the modelling mostly depends on the scale used. For large scales, such as big countries, continents and the whole world, available airport data is enough to give us reliable predictions [12]. As mentioned, [7] has some important results for the dissemination of the COVID-19 in Brazil based on airport network. But once the epidemics reaches a primary local region, it is of relevance to anticipate how the dissemination will take place locally, so local transit, commuting networks and regional road movement play an important role in the modelling [12]. However, obtaining reliable data of regional and local mobility is a great challenge, and are mostly based on census data [13, 14], which are not able to capture up-to-date mobility fluctuations and changes due to the pandemic. Fortunately, more recently, mobile geolocation data are becoming available to provide a reliable characterization of such movements, and this will be the driver of our modeling proposal.

Many studies address epidemic models on networks (e.g. [12, 13, 14, 15, 16, 17, 18, 19]). Some of these are mainly focused on using network topology information to be used as drivers for the complex system, providing relevant synthetic models that capture epidemic behaviors and produce scenarios [15]. However, in realistic metapopulation models, which is our focus, there are time dependent travel patterns and fixed populations that are difficult to address with these network models (e.g. [15, 16, 17]). In this work we focus on extracting the mobility pattern between cities (nodes of the network) directly from mobile mobility data, so we are neither assuming node degree clustering to determine its dynamic behavior, nor assuming a mean field or statistical distribution to model the dynamics. In this sense, this work is data driven and is of heuristic in nature.

Other spatial-temporal network-based models in the literature, e.g. [13, 14, 19], show metapopulation models with several compartments that rely on parameter calculations dependent not only on mobility data, obtained via census or air traffic data, but also on early information about the current disease characteristics and spreading. While these are of upmost importance

for evaluation of the long-term behavior of disease dissemination, these models bring unnecessary complexity for the evaluation of the short-term spatial spreading and brings several uncertainties to the model. In this work, we propose a simple susceptible-infected compartmental model, adequate for the initial stages of the dissemination, coupled with a rich dataset of mobile mobility data that covers more than one fourth of the Brazilian mobile devices with accuracy of meters in space and minutes in time. As a natural extension of this work, a metapopulation model with more compartments coupled with network mobility data are to be presented in a follow up paper, allowing investigation of later stages of the disease spreading.

In the first part of this study we rely on mobile data to assess the movement pattern between cities within the states of São Paulo and Rio de Janeiro in Brazil, before and during the COVID-19 pandemic, in order to identify future foci of infection within the states. We concentrate on these states as they were the first ones in Brazil to have significant number of confirmed cases and local transmission. To model the mobility via mobile data we have established a fruitful collaboration with Brazilian company In Loco (https://inloco.com.br/). In Loco provides software engineering services to mobile phone applications and has a database with more than 60 million Brazilian devices. The anonymized data provided by them contains the physical locations where billions of visits to selected apps have occurred. Although no civil information is collected, such as name or social security number, in deference to users' privacy concerns, In Loco can detect, through anonymous tracking, the most likely devices' locations across the country and the movement between them.

In this work, we measure the mobility in each day of March 2020 between the cities within each state, seeking to identify the most common mobility patterns in order to predict possible future foci of infection. We consider the movement on March 2020 as these were the days which followed the first infections in Brazil, when quarantine measures were implemented. To predict theses foci, we analyze the raw mobility data and simulate spatial-temporal models of disease spread to predict the locations where the disease is more likely to spread first. This study seeks to not only subsidize public discussions about the allocation of resources and enforcement of isolation measures, but also to be the base of a next study, addressing population dynamics together with available public health data, providing risk assessments and forecasts.

## Methods

### Dataset

The In Loco company provided anonymized data containing the geolocation of millions of users of their software development kit (SDK), which is present in many popular mobile apps. For this part of the study, we only analyze data referring to the states of São Paulo and Rio de Janeiro, although data of other states are also collected by the company. The available dataset contains, from the 1st to the 30th of March 2019 and March 2020, recordings of pairs of positions, referring to the locations of an initial and a second app use by a same device. Each position is calculated based on the location where an app with In Locos' system was used and on information collected on the background while the app was not running, which aids in the collection of data when the app is in use, and is provided in geographical coordinates with a precision of 0.01 degrees in each coordinate.

The first position refers to a use in a given day of an app by a device, while the second position refers to where a subsequent use occurred, but only when this location is different from the first one. Hence, only movements between different locations are represented, since users which used an app multiple times in a day within the same location are not present in the dataset. Furthermore, we excluded all pairs in which the second use occurred more than 24 hours

after the first one, since these represent users that were steady on a single location for over 24h, and therefore not moving within the analyzed day, or performed a movement that took over 24h time spam, without stops in between (rare). This ensures that all movements analyzed occurred inside the period of 24 hours, having started within the day under investigation. Observe that each device may appear more than once if the apps are used multiples times in a same day at different locations, although, by anonymization, we do not know how many times a device appears, hence cannot follow it for more than two consecutive uses. Therefore, we have two-point movement data in space-time of millions of devices in each day, representing a rich sample of daily population dynamics.

We will focus on mobility data from March 2019, as a reference, and March 2020, as a measure of mobility patterns during the pandemic. For São Paulo we have on average 3.6 million daily position recordings in March 2019 and 4.3 million in March 2020. For Rio de Janeiro we have on average 1 million daily recordings in March 2019 and 0.8 million in March 2020. These recordings come from approximately 10 million unique users monitored in the state of São Paulo and 3 million unique users monitored in the state of Rio de Janeiro. Just as a reference, São Paulo state has a population of approximately 45 million people and Rio de Janeiro state has approximately 16.4 million people. We note here that, even though the dataset captures only the mobility of smartphone users of a certain range of mobile applications, therefore limited in terms of socio-demographic profile, due to its large penetration in the economically active population, it tends to be a good representation of the overall population mobility patterns. In Table 1 we present descriptive statistics of the daily number of recordings for weekdays and weekends for both years. We note that the daily uses decrease on weekends, an evidence of reduced mobility between locations during weekends.

Fig 1 shows the daily number of recordings in both states in 2019 and 2020. On the one hand, in 2019 we see a steady pattern of the recordings in both states, which approximately repeats itself every week. On the other hand, in 2020, there is a clear decline in the number of recordings starting on the 15th, especially in Rio de Janeiro. This decline coincides with the implementation of stronger isolation measures enforced on the second half of March. Indeed, in Fig 2 we see a great decrease on the number of recordings in the second half of March (starting on the 15th), in both weekends and weekdays, as the boxes, which illustrate the statistics in Table 1, are below the respective boxes in the first half of March. As the control group (March 2019) behaves approximately the same on the first and second half, showing evidences that the isolation measures implemented decreased the mobility in these states. Now, since the dataset contains only recordings of movement, the number of recordings is, by itself, an intrinsic measure of population isolation/quarantine, hence its decline is an evidence of efficacy of isolation measures, as observed in other countries (e.g. France [20]).

Table 1. Descriptive statistics of the daily number of recordings in March 2019 and 2020 for each state on the weekends and weekdays.

| State | Year | Day Week | Mean | SD | Min | 1st Quart. | Median | 3rd Quart. | Max |
|---|---|---|---|---|---|---|---|---|---|
| RJ | 2019 | Weekday | 1,053,615 | 259,721 | 528,805 | 790,609 | 1,140,444 | 1,180,351 | 1,465,666 |
| | | Weekend | 938,472 | 163,672 | 679,678 | 811,883 | 962,988 | 1,026,558 | 1,201,777 |
| | 2020 | Weekday | 870,920 | 445,958 | 214,521 | 509,431 | 870,189 | 1,196,752 | 1,682,386 |
| | | Weekend | 624,417 | 425,258 | 179,309 | 238,466 | 442,211 | 1,023,705 | 1,332,701 |
| SP | 2019 | Weekday | 3,708,276 | 850,839 | 2,221,510 | 2,790,522 | 3,927,577 | 4,169,805 | 5,011,449 |
| | | Weekend | 3,256,169 | 563,222 | 2,456,231 | 2,756,478 | 3,550,141 | 3,569,211 | 4,172,801 |
| | 2020 | Weekday | 4,353,782 | 1,652,625 | 1,661,284 | 2,816,090 | 4,465,741 | 5,545,852 | 7,384,012 |
| | | Weekend | 3,561,949 | 1,495,902 | 1,681,135 | 2,118,311 | 3,964,115 | 4,999,465 | 5,527,734 |

SD = Standard Deviation.

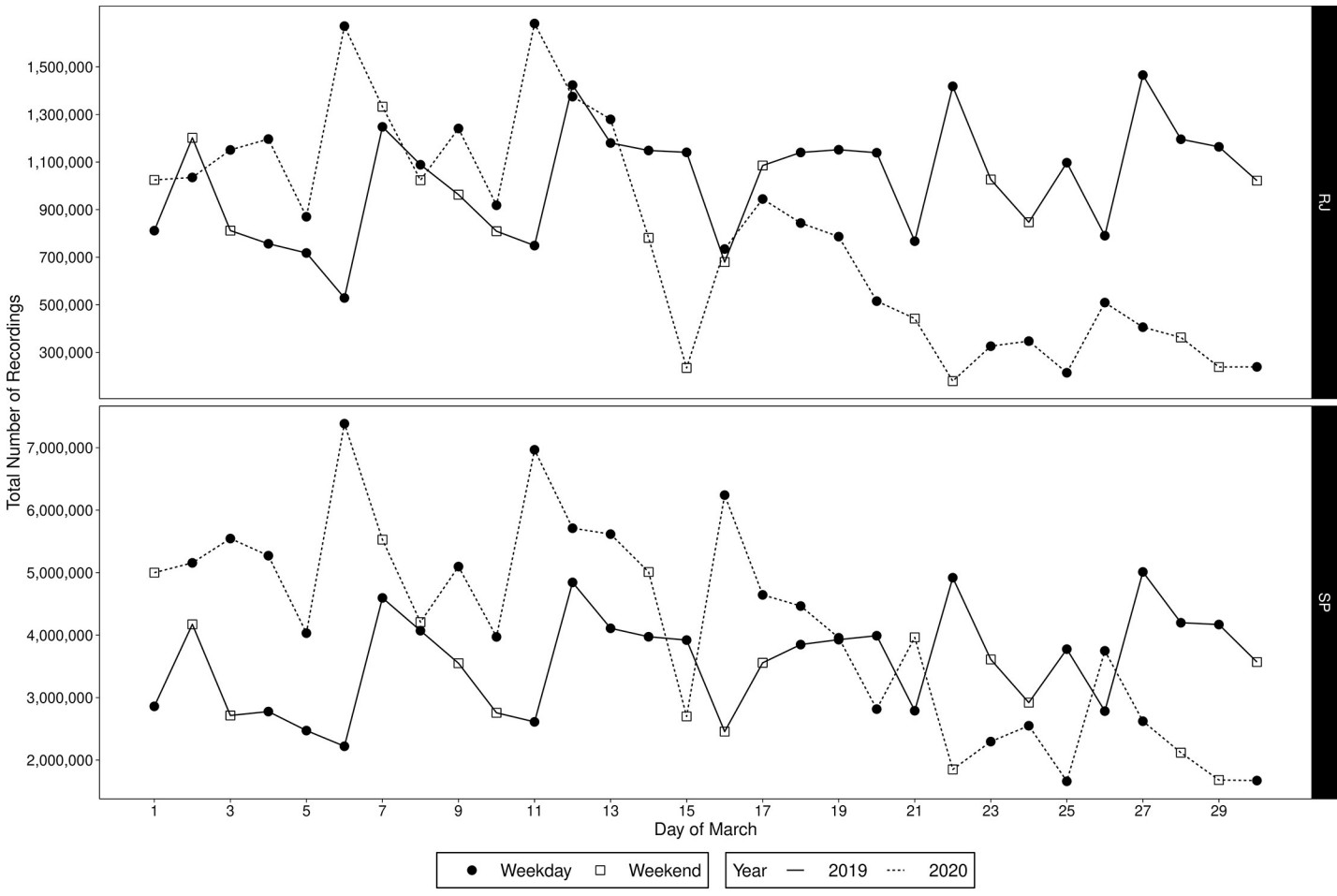

**Fig 1. Total number of recordings for each day of March in São Paulo (SP) and Rio de Janeiro (RJ), in 2019 and 2020.**

Fig 3 shows the number of recordings in each location in a usual day of March. There is a clear pattern on the distribution of these locations, which are concentrated within cities and along roadways, in both states. Furthermore, the majority of uses occurred in the surroundings of the states' capital cities, in their metropolitan region. The pattern of these locations evidences how this data is a good proxy for population mobility, as it is either representing movement within cities or between them, via roadways. This distribution of locations is a good evidence in support of mobile data to assess regional mobility.

## Movement dynamics

In order to study mobility patterns between cities we group the recordings by city, i.e., each position is mapped from geographical position to the city containing it, generating a sample with pairs of initial city and subsequent city, according to the movement given by the geolocation. If the two positions are within the same city, we consider that there has been no movement, as *movement* here is taken as *movement between cities*. Proceeding in this manner, we divided São Paulo in 645 regions and Rio de Janeiro in 92 regions, given by their cities. Although we chose to divide the states by cities, we could have chosen another division, with more or less resolution, considering for example microregions (formed by cities) or

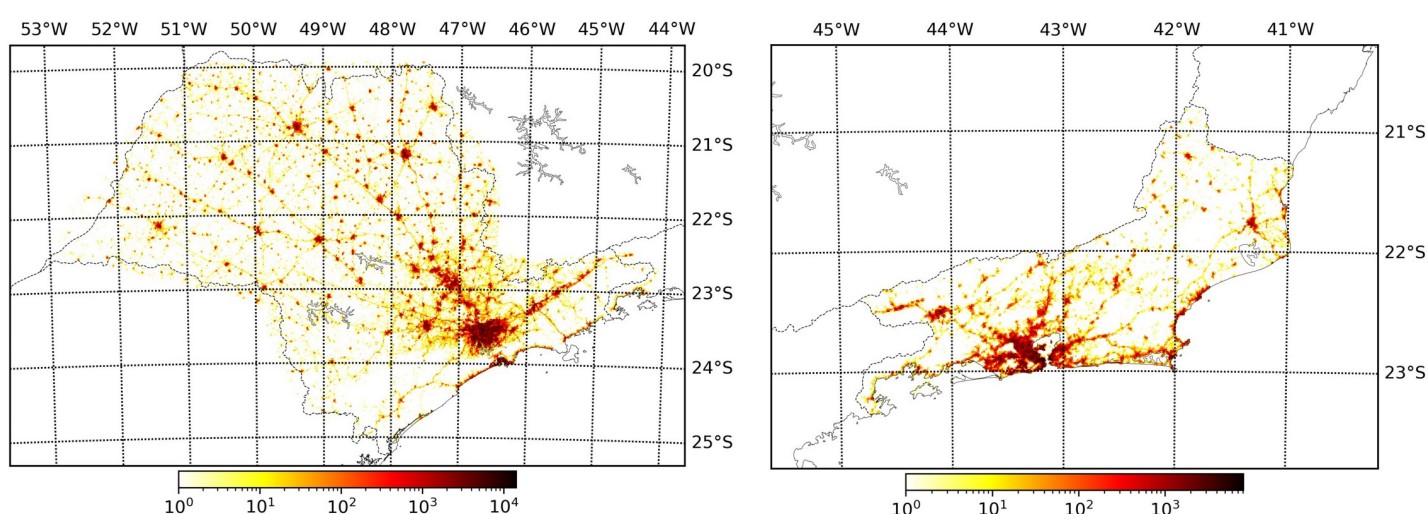

**Fig 2. Box-plot of the total number of recordings on the first (from the 1st to the 14th) and second (from the 15th to the 30th) half of March, in weekdays and weekends, in 2019 and 2020 for both states.**

**Fig 3. Typical distribution of the location of app usage in one day for the states of São Paulo (left) and Rio de Janeiro (right) considering a resolution of 0.01 degree on each geographical coordinate.** This data refers to March 1st, 2020 and the color represents the number of recordings, first or subsequent, in each location.

subdistricts (which form cities), in order to study the dynamics in larger or smaller scales. The city aggregation seems to provide a good trade-off between data availability and the population density distribution within the states, as higher resolution divisions leads to regions with relatively small samples of mobile users from the data base, and lower resolutions lead to loss of important mobility patterns responsible for the disease dissemination.

From the generated sample of movements between regions, we can compute the proportion of movements from a region A to each other region in a period of time. In this study, we always consider the period to be that of a day. This proportion of movement from region A to a region B is given simply by the number of recordings which departed from A in the given day and were in B within 24 hours, divided by the total number of recordings which started in A in the given day. This is the proportion of movements starting from A in a day and ending in B within 24 hours. Also, the proportion of *no-movement* of A in a day is given by the number of recordings which started in A in this day and were still in A in the second use, less than 24 hours later. These proportions are organized in a transition matrix, in which the entry of column A and row B is the proportion of movement from A to B. For each considered day in March 2019 and 2020, we have a transition matrix containing the movement between regions in this day.

As we lack information about consecutive uses which occurred in a same location, the proportion of no-movement of a region is underestimated. All self-loop movements relate only to movements within the city, disregarding devices which have not moved at all within a day. This causes the proportion of movement from A to other regions to be overestimated. For example, we may observe proportions as high as 45% of a city population moving outwards it in a day, which is unrealistic. This can be solved using additional data about the users that did not move on the given day in the proportion calculation. Unfortunately this information is not available from the given dataset, therefore this overestimation will be corrected with an additional parameter in the models simulating the disease spread. However, when analyzing raw data we disregard any correction, as we are only interested in determining common movements patterns, with its relative intensities only.

Even though this proportion is not a consistent estimator, in a statistical sense, of the proportion of a population which travels from a region to another within 24 hours, as a same device may be recorded twice in the period of a day, it is a good proxy for the mobility between two regions. As the data is anonymized and each device is followed for only two uses, we do not actually know if the movement is that of a person which is returning to a location or going there for the first time, for example. However, this proportion gives a good idea of possible patterns followed by a population in general, since a recurrent movement pattern in the population will also be present in our dataset, although the proportion of movements in the population may be of distinct intensity compared to the one we calculated.

In order to assess the mobility patterns in weeks following the first cases of COVID-19 in Brazil in March 2020, we always take the mobility in March 2019 as a control group. Indeed, we need a measure of the usual mobility between the regions to compare with the observed mobility to know if it is within the usual pattern. For this purpose, we disregard the first days of March 2019, as the mobility was influenced by a major Brazilian holiday, the carnival week, so we observe the pattern of mobility in March 2019 starting on the 11th. On the one hand, the mobility in March 2020 is measured daily, by the proportion of movement from one region to another, i.e., by the daily transition matrices. On the other hand, the mobility in March 2019 is measured by the mean of these proportions over all considered days of March which fell on a day of the week, i.e., for each day of the week we calculate the mean of the proportions for all considered days of March 2019 which fell on it. Proceeding in this way, we have one transition matrix for each day of March 2020, and seven transition matrices related to the mean pattern of movement of each day of the week in March 2019. Each day of March 2020 is compared with the pattern of the day of the week it fell on. The analysis

of this study concentrates on an important feature of the pandemic spread, that is, possible focus of future infections. We now discuss how they can be evaluated from the available mobile data.

## Possible foci of infection

The COVID-19 outbreak in the states of São Paulo and Rio de Janeiro has started in their capital city in the end of February 2020 and spread to other cities on the metropolitan region and countryside. However, many regions were yet to suffer from the pandemic, so pointing out possible focus of future infections provides strategic information to public authorities. These foci may be identified by studying the pattern of movement from the infected regions (capital cities) to the countryside, by identifying common movement patterns. Observe that the geographical distance between cities is not enough to determine these foci, as there are other factors which drive mobility within the states, specially of economic nature, which make movement to more developed cities far away more likely.

The analysis is focused on the movement patterns starting from the capital cities and is performed with the aid of maps, in which each region is painted according to the proportion of the movements from the capital city which ended in each region. The daily patterns in these movements in March 2020 provide insights about possible paths infected people may have taken, spreading the disease to other regions. Also, we study the most frequent movements from the capitals in the days of March 2020 and March 2019 seeking to find common movements, and any difference in the patterns, from one year to the next.

## A model for the spatial spreading of the disease

The main focus of this study is to explore the mobility dynamics within a state in order to give authorities a heads up on the evolution of the disease, so they can be a step ahead and prepare the local health care systems for the upcoming events. Since we do not have reliable data on the recovery time, we decided to use in this first approach an infectious model suitable for the initial exponential spread of the disease. Once we have more reliable data, we can incorporate other nuances of the disease spread and infection to get more adequate models for the next stages of the spread. So, in order to model the spatial spread of the disease in this early stage, we consider a metapopulation model which relates the evolution of a disease inside a population with two terms, one referring to the spread within the location and another to the spread to and from other locations. The spread within each location is modelled as a SI model, while the spread between locations is based on the mobile data, more specifically on the transition matrices. In the proposed model, the evolution of $I_i(t)$, the number of infected in region $i$ at time $t$, is modelled as

$$\frac{dI_i}{dt}(t) = (1 + r)I_i(t)\left(\frac{N_i - I_i(t)}{N_i}\right) + s\left[\sum_{j \neq i} \omega_{ji}(t)I_j(t) - \sum_{j \neq i} \omega_{ij}(t)I_i(t)\right]$$

in which $r$ is the transmission rate within each region, $s$ is a free parameter used to correct the overestimation or underestimation of movement between the locations, $N_i$ is the population of region $i$ and $\omega_{ji}(t)$ is a measure of the movement from region $j$ to region $i$ at time $t$, calculated from the transition matrices in the following way.

Let $\hat{p}_{ji}^n$ be the entry at row $j$ and column $j$ of the transition matrix of day $n$, indicating the proportion of registered movements from region $j$ to $i$ at day $n$, and let $m_{ji}^n = \hat{p}_{ji}^n R_j^n$, where $R_j^n$ is the total number of recordings which departed from $j$ at day $n$, resulting in the actual number of recorded movements from region $j$ to $i$ in this day. We take $m_{ji}^n$ as an estimative of the number of people which moved from region $j$ to region $i$ at the day. We consider the measure of

mobility from $j$ to $i$ as

$$\omega_{ji}(t) = \frac{m_{ji}^n}{N_j}$$

in which $n$ is such that $24(n-1) \leq t < 24n$. The scale of $t$ that we consider is that of an hour, and $t = 0$ is midnight at March $1^{st}$ 2020. Also, we consider $r = 0.4$ which is approximately $R_0/6$, in which $R_0 = 2.68$ is the Basic Reproduction Number estimated by [21] from data about the disease spread in Wuhan, China, and 6 days is the mean incubation time of the disease.

Although the SI model is not suitable for long forecasts, since it does not considers the Recovered and Exposed individuals, we explore it simulating the spread until the end of April 2020. But, as there is no mobility data available beyond March $30^{th}$, 2020, we will use the mean transition matrices from the corresponding weekday in March 2020 when simulating the spread in April. At $t = 0$ we start with one single infected case in the state's capital and zero in the other cities, and simulate how the disease spreads spatially within the states.

We use the number of recordings from one region to another divided by the population of the departure region as the measure of mobility between regions because the proportion of transitions may overestimate the mobility, as we do not have data about devices which have not moved within a day. When dividing by the population, we may assume that the number of recordings is actually the number of people moving from one region to another. However, this estimator is biased, on the one hand, as each device may be counted more than once, and, on the other hand, there are people moving between regions without using any app. Therefore, we need to correct the estimative, which is performed using and additional parameter $s$, multiplying the proportions. If $s < 1$, then we are correcting for a possible overestimation of the movement proportions, while if $s > 1$, we are correcting a possible underestimation of the proportions. Hence, we will simulate the model for various values of $s$ to attest its robustness.

The main interest of the simulations is in determining $t_i$, the least time such that the number of infected in a region $i$ attains a threshold $c$, i.e., $I_i(t) \geq c$. From this value, we may rank the regions from the smallest to the greatest times of arrival of the disease, producing evidences about possible foci of future infection. In the simulations we adopt $c = 1$, that is, we assume that the region is at risk when the model predicts at least 1 infected individual in the region. The models are simulated until April $30^{th}$, 2020.

## Results

### Possible foci of infection

In Figs 4 and 5 we present the proportions of movements from the capital cities at March $1^{st}$, $10^{th}$, $20^{th}$ and $30^{th}$ 2020, and the mean proportions of the respective day of the week from March 2019. We observe that the mobility pattern is similar in both years, although the value of the proportions may differ. As we have seen, the number of recordings decreased in the second half of March 2020 influenced by quarantine measures, but according to Figs 4 and 5 the movement patterns did not change significantly. This means that, among people still moving between cities, the pattern is that of before social distancing, hence distancing measures seems to have not changed the pattern movement, at least in the city scale, but only the intensity of movement, evidenced by the decrease on the number of recordings.

In Tables 2 and 3 we show descriptive statistics of the rank of the top 15 cities concentrating the proportion of movement out of the capitals, calculated for all days of March 2019 and 2020. The rank is the ordering, from lowest to greatest, of the proportion of movement from the capital, so as greater the rank, more movement was observed from the capital to the city. We see that the rank does not vary much among the days of March (small standard deviation), and that the rank

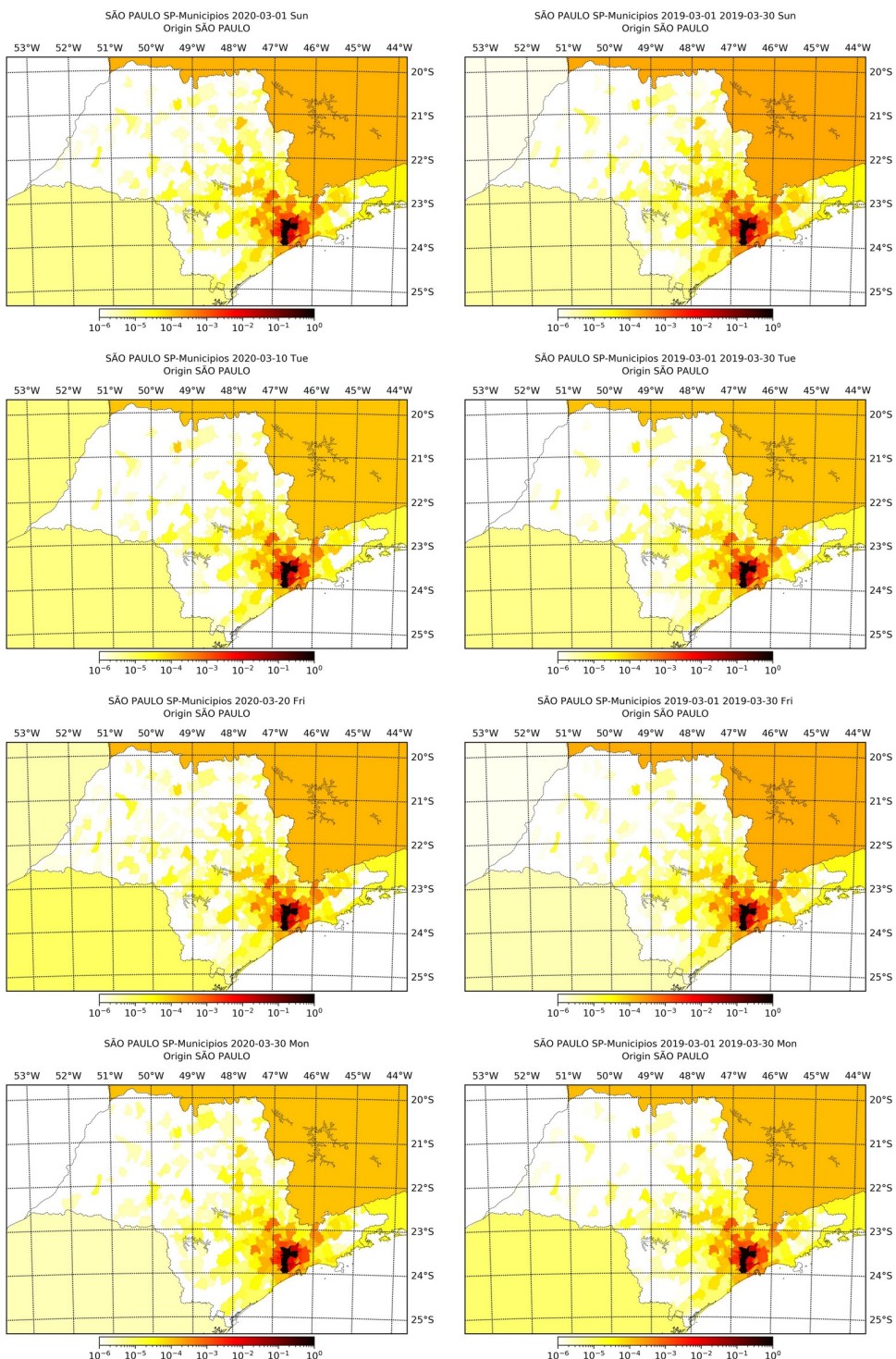

**Fig 4. Proportion of movement from São Paulo capital city to each city within the state at March 1st, 10th, 20th and 30th of 2020 alongside with the mean proportion of movement of the respective weekday in March 2019.**

in 2019 is close to the rank in 2020, evidencing again that, even though movement has decreased, the pattern of movement has not changed. These top cities are mainly in the metropolitan region of the capitals, an evidence that these may be future foci of infection.

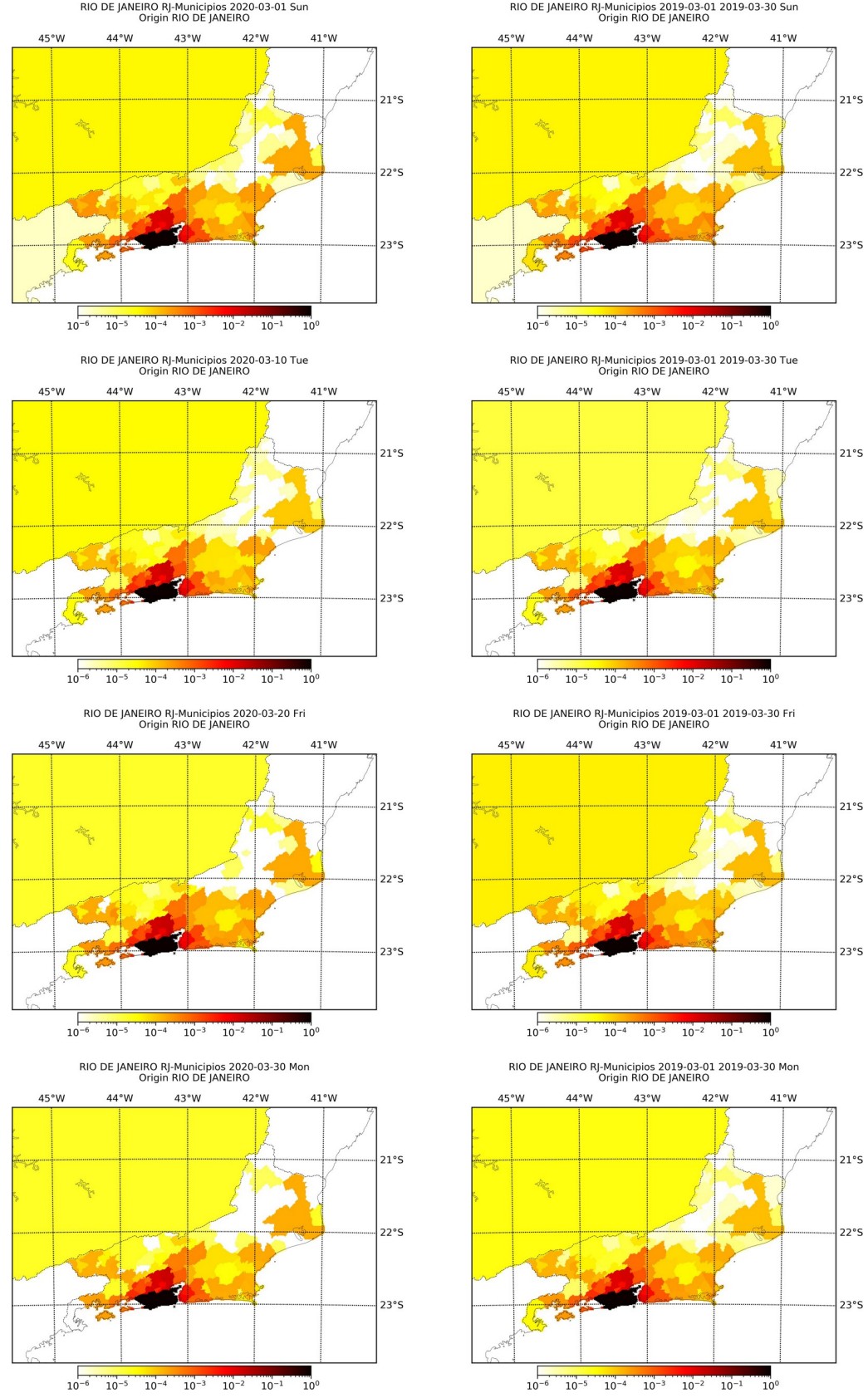

**Fig 5. Proportion of movement from Rio de Janeiro capital city to each city within the state at March 1st, 10th, 20th and 30th of 2020 alongside with the mean proportion of movement of the respective weekday in March 2019.**

## Model for spatial disease spreading

Figs 6–9 display the simulated number of infected individuals across the states of São Paulo and Rio de Janeiro as time evolved for selected values of $s$. We see that the effect of $s$ is on the time, in number of days, that the disease takes to attain some location, rather than on the evolution of the spread itself. For $s = 1$, we observe in Figs 6 and 8 that the number of infected individuals spread from the capital cities, to the their metropolitan region and then selected cities on the countryside, which are geographically far from the capital, especially in the state of São Paulo. We see in Figs 7 and 9 that, for different values of $s$, the evolution of the disease is the same, but the cities with lager infection counts at the end of the simulation, i.e., April 30th, depend on $s$: the larger the value of $s$, more cities are infected at the end. Also, we can clearly see a non-local diffusion process, as described by [9].

**Table 2. Descriptive statistics of the rank of the proportion of movement out of São Paulo capital city in the days of March 2019 and March 2020.**

| City | Year | Mean | SD | Min | 1st Quart. | Median | 3rd Quart. | Max |
|---|---|---|---|---|---|---|---|---|
| Guarulhos | 2019 | 648 | - | 648 | 648 | 648 | 648 | 648 |
| Guarulhos | 2020 | 648 | - | 648 | 648 | 648 | 648 | 648 |
| Osasco | 2019 | 647 | - | 647 | 647 | 647 | 647 | 647 |
| Osasco | 2020 | 647 | - | 647 | 647 | 647 | 647 | 647 |
| Santo André | 2020 | 646 | - | 646 | 646 | 646 | 646 | 646 |
| Santo André | 2019 | 646 | 0,30 | 645 | 646 | 646 | 646 | 646 |
| Taboão da Serra | 2019 | 645 | 0,30 | 645 | 645 | 645 | 645 | 646 |
| Taboão da Serra | 2020 | 645 | - | 645 | 645 | 645 | 645 | 645 |
| Diadema | 2019 | 644 | - | 644 | 644 | 644 | 644 | 644 |
| Diadema | 2020 | 644 | - | 644 | 644 | 644 | 644 | 644 |
| S. Bernardo do Campo | 2019 | 643 | - | 643 | 643 | 643 | 643 | 643 |
| S. Bernardo do Campo | 2020 | 643 | - | 643 | 643 | 643 | 643 | 643 |
| São Caetano do Sul | 2019 | 642 | 0,40 | 641 | 642 | 642 | 642 | 642 |
| Barueri | 2020 | 642 | 1,53 | 637 | 642 | 642 | 642 | 642 |
| Embu das Artes | 2019 | 641 | 0,48 | 640 | 641 | 641 | 641 | 642 |
| São Caetano do Sul | 2020 | 641 | 0,40 | 639 | 641 | 641 | 641 | 641 |
| Embu das Artes | 2020 | 640 | 0,61 | 640 | 640 | 640 | 640 | 642 |
| Itaquaquecetuba | 2019 | 640 | 0,65 | 638 | 640 | 640 | 640 | 640 |
| Itaquaquecetuba | 2020 | 639 | 0,53 | 639 | 639 | 639 | 639 | 641 |
| Ferraz de Vasconselos | 2019 | 639 | 0,82 | 636 | 639 | 639 | 639 | 640 |
| Mauá | 2019 | 638 | 0,54 | 638 | 638 | 638 | 638 | 640 |
| Mauá | 2020 | 638 | - | 638 | 638 | 638 | 638 | 638 |
| Barueri | 2019 | 638 | 1,65 | 636 | 637 | 637 | 637 | 642 |
| Ferraz de Vasconselos | 2020 | 637 | 0,73 | 637 | 637 | 637 | 637 | 640 |
| Carapicuíba | 2020 | 636 | 0,25 | 635 | 636 | 636 | 636 | 636 |
| Itapecerica da Serra | 2019 | 636 | 0,65 | 634 | 636 | 636 | 636 | 637 |
| Carapicuíba | 2019 | 635 | 0,45 | 635 | 635 | 635 | 635 | 637 |
| Itapecerica da Serra | 2020 | 635 | 0,27 | 635 | 635 | 635 | 635 | 636 |
| Cotia | 2019 | 634 | 0,30 | 634 | 634 | 634 | 634 | 635 |
| Cotia | 2020 | 634 | 0,09 | 634 | 634 | 634 | 634 | 635 |

**Table 3. Descriptive statistics of the rank of the proportion of movement out of Rio de Janeiro capital city in the days of March 2019 and March 2020.**

| City | Year | Mean | SD | Min | 1st Quart. | Median | 3rd Quart. | Max |
|---|---|---|---|---|---|---|---|---|
| Duque de Caxias | 2019 | 94 | - | 94 | 94 | 94 | 94 | 94 |
| Duque de Caxias | 2020 | 94 | - | 94 | 94 | 94 | 94 | 94 |
| Nova Iguaçu | 2020 | 93 | - | 93 | 93 | 93 | 93 | 93 |
| São João de Meriti | 2019 | 93 | 0,43 | 92 | 93 | 93 | 93 | 93 |
| Nova Iguaçu | 2019 | 92 | 0,43 | 92 | 92 | 92 | 92 | 93 |
| São João de Meriti | 2020 | 92 | - | 92 | 92 | 92 | 92 | 92 |
| Niterói | 2019 | 91 | - | 91 | 91 | 91 | 91 | 91 |
| Niterói | 2020 | 91 | - | 91 | 91 | 91 | 91 | 91 |
| Belford Roxo | 2020 | 90 | 0,25 | 89 | 90 | 90 | 90 | 90 |
| Nilópolis | 2019 | 90 | 0,51 | 88 | 90 | 90 | 90 | 90 |
| Belford Roxo | 2019 | 89 | 0,52 | 88 | 89 | 89 | 89 | 90 |
| São Gonçalo | 2020 | 89 | 0,37 | 88 | 89 | 89 | 89 | 90 |
| São Gonçalo | 2019 | 88 | 0,63 | 88 | 88 | 88 | 88 | 90 |
| Nilópolis | 2020 | 88 | 0,43 | 88 | 88 | 88 | 88 | 90 |
| Itaguaí | 2020 | 87 | 0,31 | 86 | 87 | 87 | 87 | 87 |
| Mesquita | 2019 | 87 | 1,03 | 83 | 87 | 87 | 87 | 87 |
| Mesquita | 2020 | 86 | 0,31 | 86 | 86 | 86 | 86 | 87 |
| Itaguaí | 2019 | 86 | 0,47 | 85 | 86 | 86 | 86 | 87 |
| Queimados | 2020 | 85 | 0,18 | 84 | 85 | 85 | 85 | 85 |
| Queimados | 2019 | 84 | 0,97 | 80 | 84 | 84 | 84 | 85 |
| Seropédica | 2020 | 84 | 0,74 | 82 | 84 | 84 | 84 | 85 |
| Itaboraí | 2020 | 83 | 0,69 | 81 | 83 | 83 | 83 | 84 |
| Seropédica | 2019 | 82 | 1,00 | 80 | 83 | 83 | 83 | 83 |
| Magé | 2020 | 82 | 0,43 | 82 | 82 | 82 | 82 | 83 |
| Magé | 2019 | 82 | 0,52 | 80 | 82 | 82 | 82 | 83 |
| Maricá | 2019 | 81 | 1,27 | 80 | 81 | 81 | 81 | 86 |
| Maricá | 2020 | 81 | 0,25 | 81 | 81 | 81 | 81 | 82 |
| Itaboraí | 2019 | 81 | 1,50 | 80 | 80 | 80 | 80 | 85 |
| Petrópolis | 2020 | 80 | 0,35 | 79 | 80 | 80 | 80 | 80 |
| Petrópolis | 2019 | 79 | 0,40 | 77 | 79 | 79 | 79 | 79 |

In order to evaluate the risk of infection of each city we consider the *rank of infection* obtained by the simulated models, as follows. For each value of *s* we number the cities by the order of disease arrival. The first city in which it arrives we rank as one, the second as two, and so forth. If the disease arrives at more than one city at a same day, they receive the same rank, and the next city in which the disease arrives receives the following rank, independently of how many cities got the disease before it. We have then for each value of $s \in$ {0.001,0.005,0.1,0.2,0.4,0.5,0.6,0.8,1,1.2,1.4,1.6,1.8,2,2.5,3} a rank for each city. The risk of infection is then calculated via a cluster analysis, in the following way.

We apply k-means clustering [22] to divide the cities into three groups (low risk, medium risk and high risk) according to their ranks attributed by the models. We first clustered the cities according to the ranks attributed by the models simulated with the values of *s* lesser than one, and for the values greater or equal to one, separately. As the clustering by both methods was very similar, as they classified differently only 49 cities in São Paulo and 29 in Rio de Janeiro, we decided to consider the ranks attributed by all values of *s* together to cluster the cities. The risk class of each city in the states of São Paulo and Rio de Janeiro is represented in the

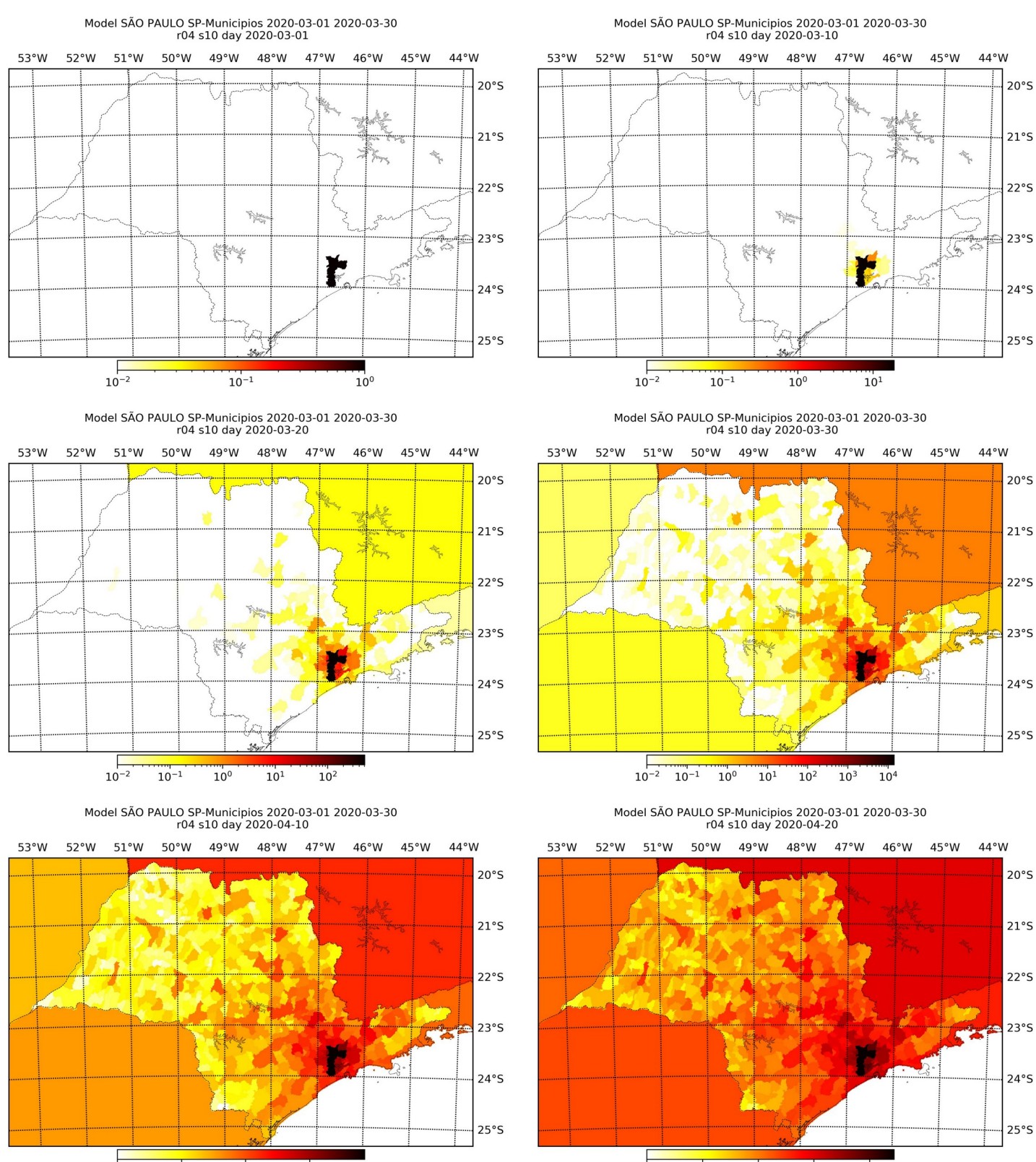

**Fig 6. Simulation results for the number of infected individuals in the state of São Paulo for selected days assuming _s_ = 1.** The maps refer, respectively, to March 1st, 10th, 20th and 30th, and April 10th and 20th.

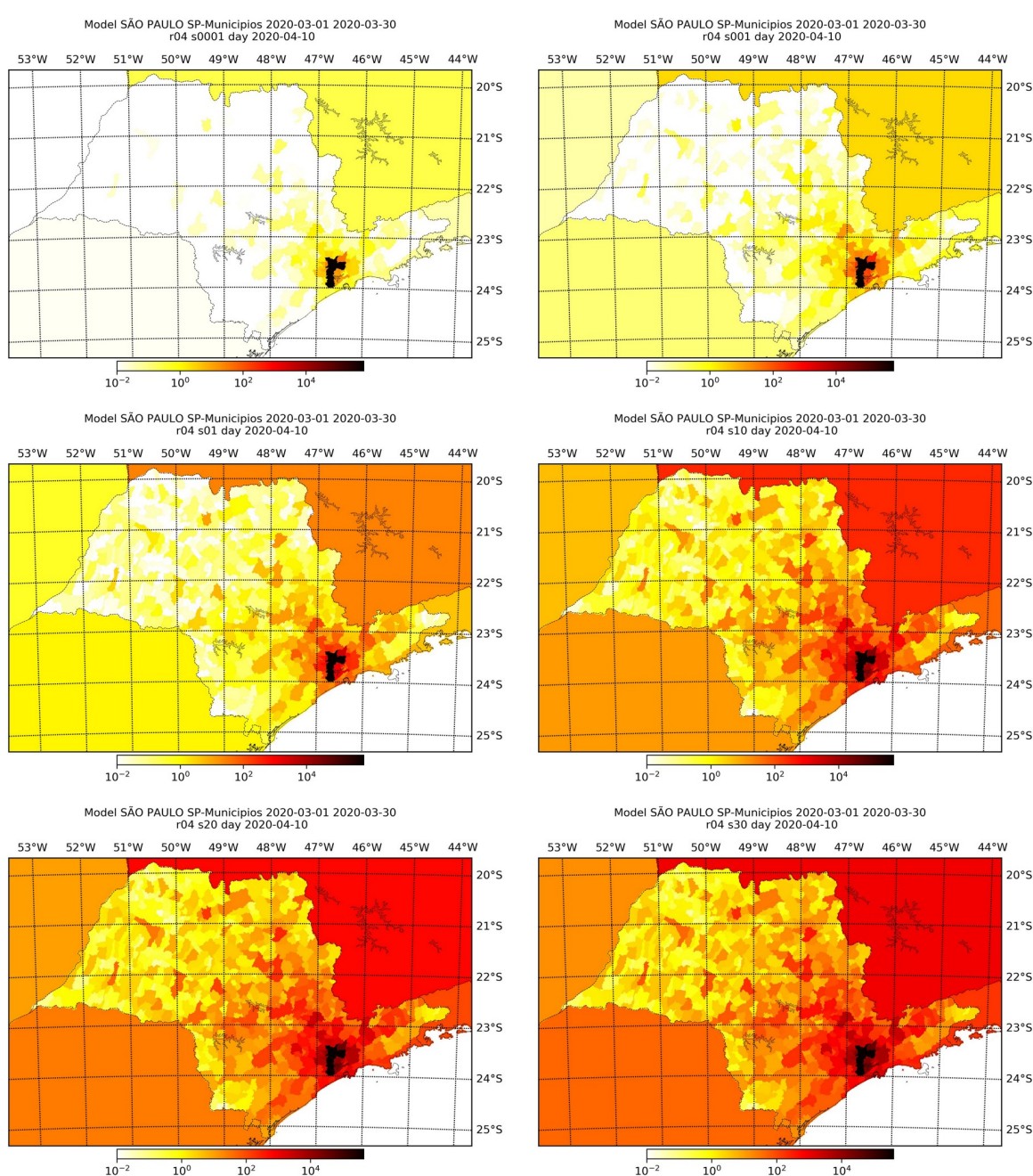

**Fig 7.** Simulation results at the 10th of April for the number of infected individuals in the state of São Paulo considering different values of *s*, namely, 0.0001 (top left), 0.001 (top right), 0.1 (mid-left), 1.0 (mid-right), 2.0 (bottom-left) and 3.0 (bottom-right).

maps of Figs 10 and 11, respectively. We see that, besides some cities in the countryside in the state of São Paulo, the high-risk locations are indeed in the metropolitan region of the capitals.

In Figs 12 and 13 we present the rank attributed by the simulated models, and the distance to the capital city, for each city with more than 100,000 inhabitants in São Paulo and more than 75,000 inhabitants in Rio de Janeiro. We observe that the rank does not change significantly with the value of *s* and see that there is a correlation between the rank of the city and the distance from the capital, since as lower the rank is, lower tends to be the distance. These

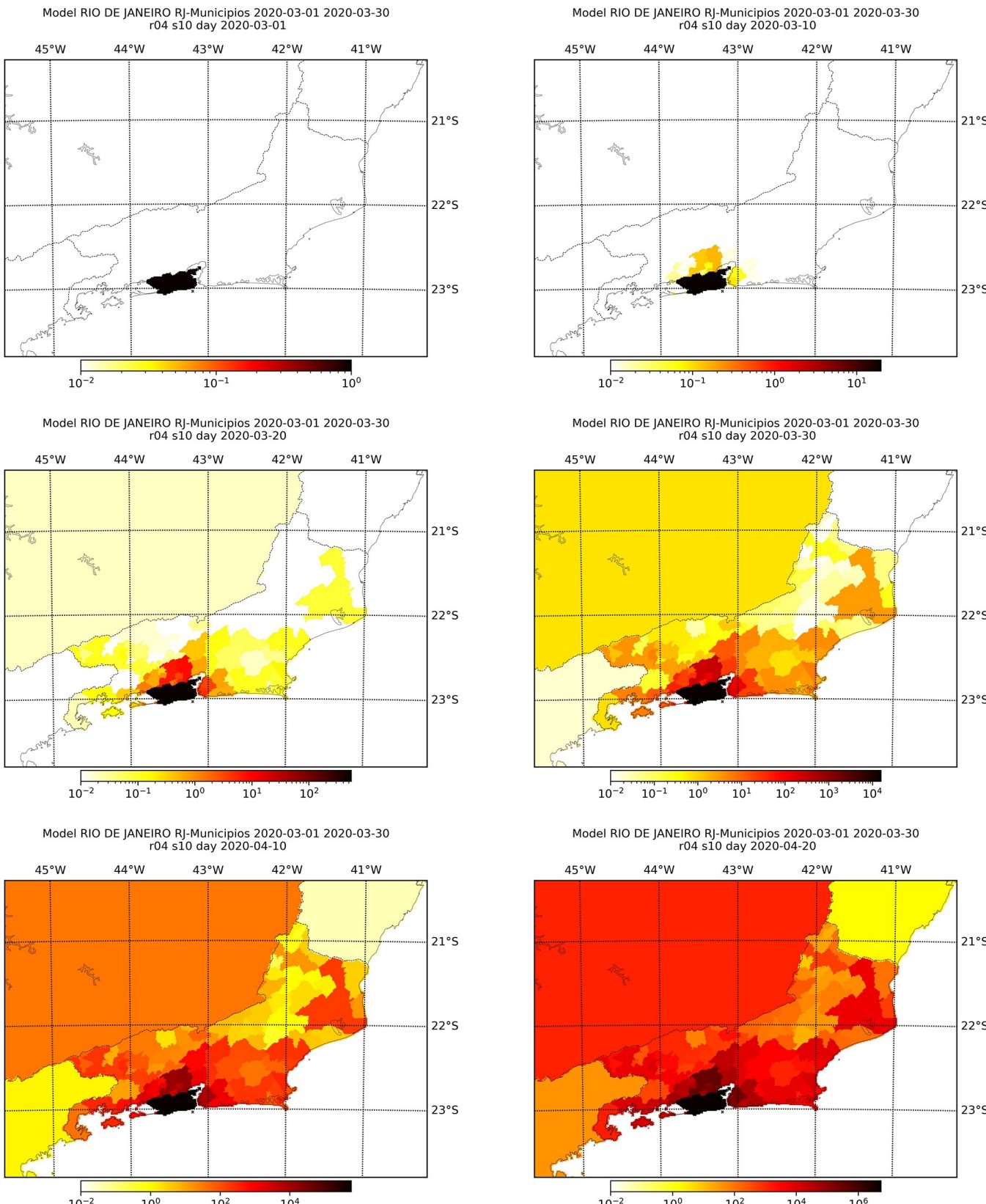

**Fig 8. Simulation results for the number of infected individuals in the state of Rio de Janeiro for selected days assuming $s = 1$.** The maps refer, respectively, to March 1st, 10th, 20th and 30th, and April 10th and 20th.

Model RIO DE JANEIRO RJ-Municipios 2020-03-01 2020-03-30
r04 s0001 day 2020-04-10

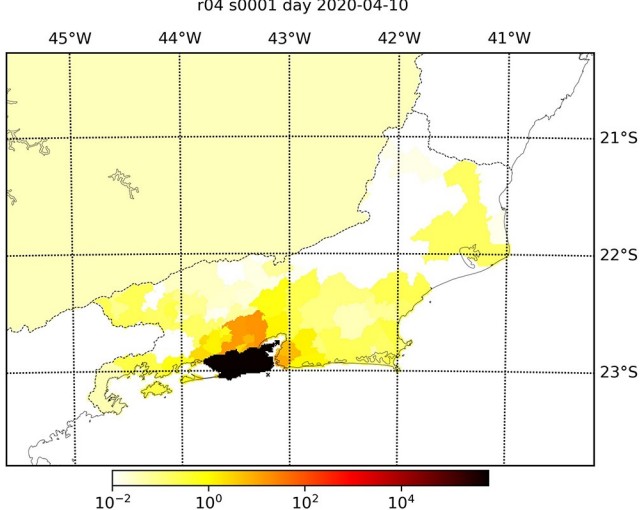

Model RIO DE JANEIRO RJ-Municipios 2020-03-01 2020-03-30
r04 s001 day 2020-04-10

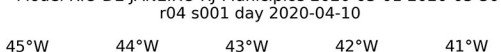
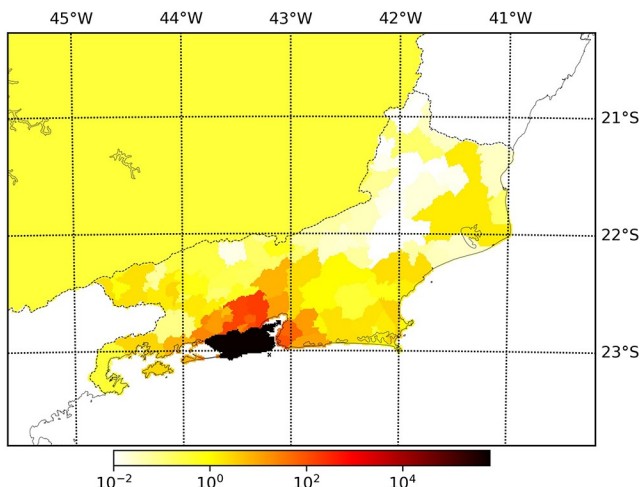

Model RIO DE JANEIRO RJ-Municipios 2020-03-01 2020-03-30
r04 s01 day 2020-04-10

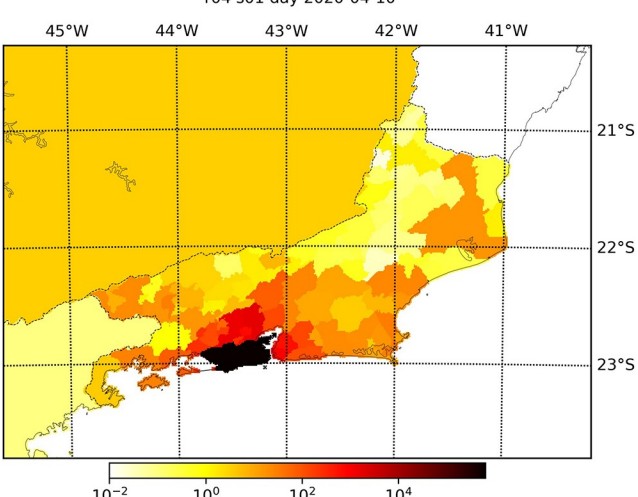

Model RIO DE JANEIRO RJ-Municipios 2020-03-01 2020-03-30
r04 s10 day 2020-04-10

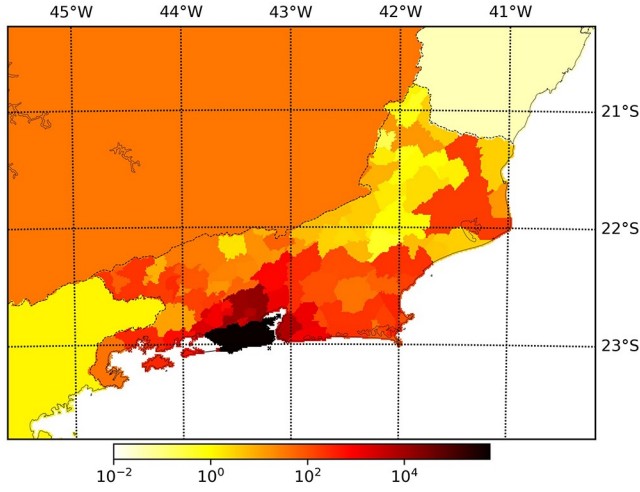

Model RIO DE JANEIRO RJ-Municipios 2020-03-01 2020-03-30
r04 s20 day 2020-04-10

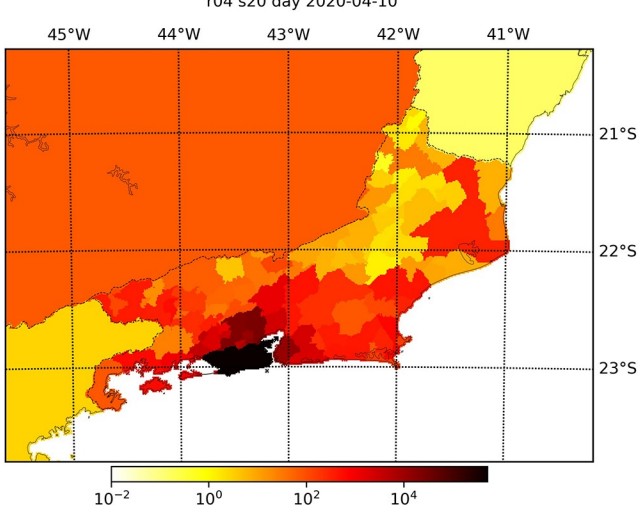

Model RIO DE JANEIRO RJ-Municipios 2020-03-01 2020-03-30
r04 s30 day 2020-04-10

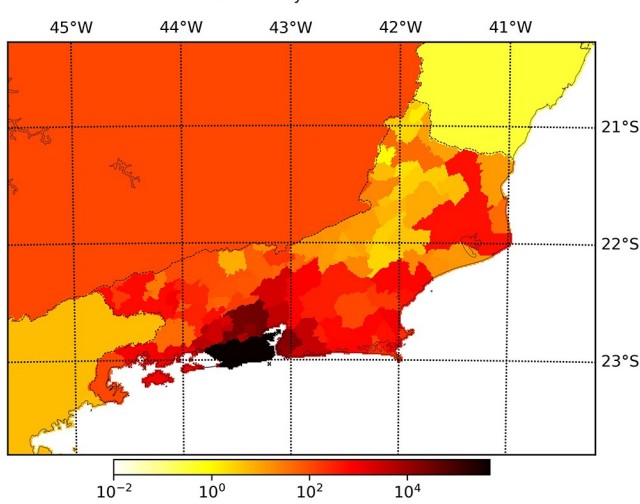

**Fig 9.** Simulation results at the 10[th] of April for the number of infected individuals in the state of Rio de Janeiro considering different values of *s*, namely, 0.0001 (top-left), 0.001 (top-right), 0.1 (mid-left), 1.0 (mid-right), 2.0 (bottom-left) and 3.0 (bottom-right).

figures show that the model, when used to predict where the disease will arrive first, is robust regarding the values of *s*, as distinct values of *s* generated similar ranks.

## Discussion

In this work, we used anonymized mobile phone data to detect population movement between cities. This framework is useful for a variety of applications. Here we focused in establishing a risk map for the evolution of the COVID-19 within the states of São Paulo and Rio de Janeiro, and noted that the high risk regions are mainly in the metropolitan region of the states' capital cities, although there are some high risk cities in the countryside, especially in São Paulo. This was done by coupling the predicted mobility patterns with a standard SI model via a metapopulation model.

The SI model is not suited for predicting the incidence of the infection for long periods of time, but it is an adequate linear approximation for the early exponential spread. The model chosen was adequate to be used with the available disease information, namely, the Basic Reproduction Number $R_0$ estimated from the initial spread in China. We also introduced *s*, a free parameter, used to correct the overestimation or underestimation of movement between

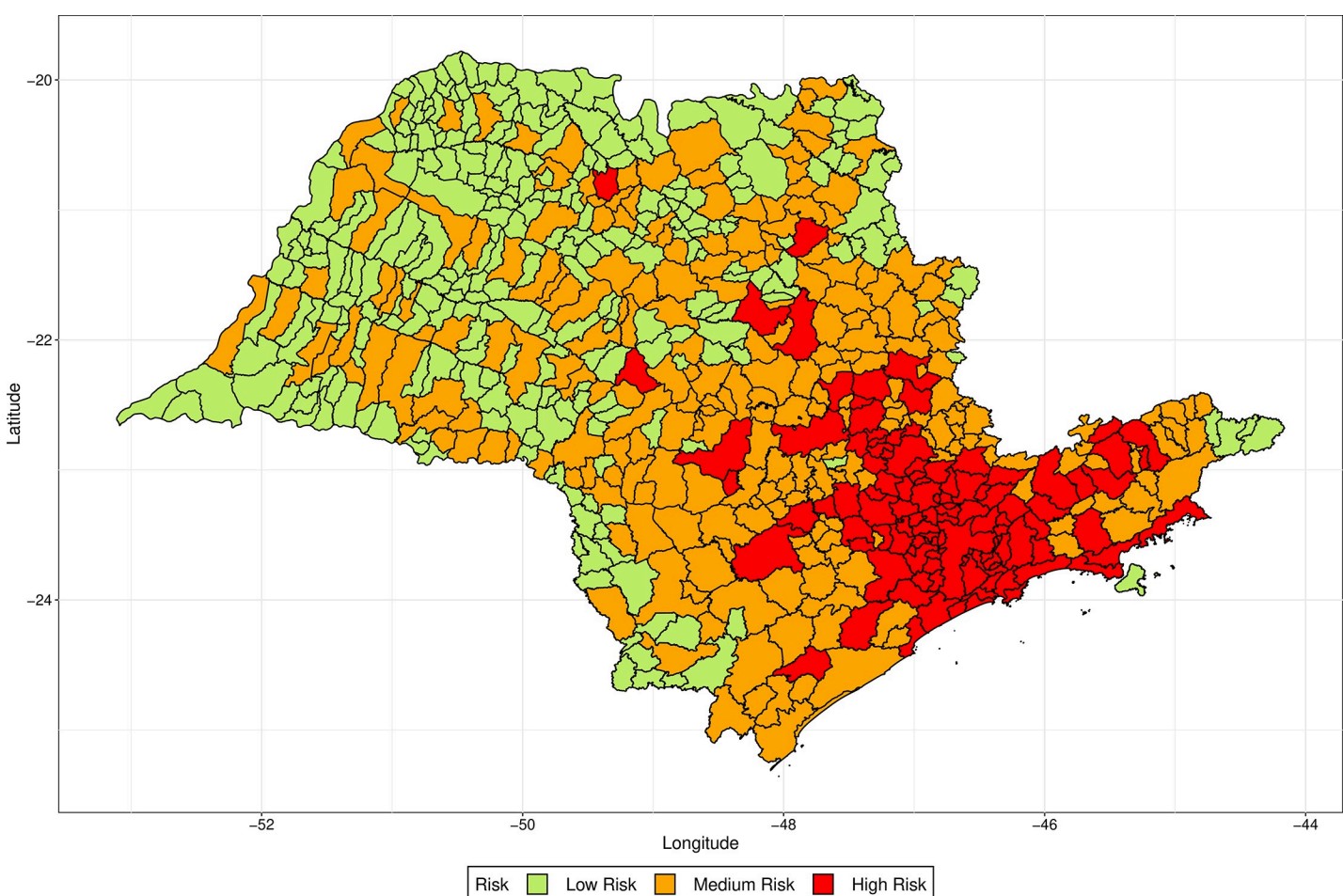

**Fig 10. Risk of each city in the state of São Paulo evaluated by k-means clustering of the ranks attributed by the simulated models with distinct values of *s*.**

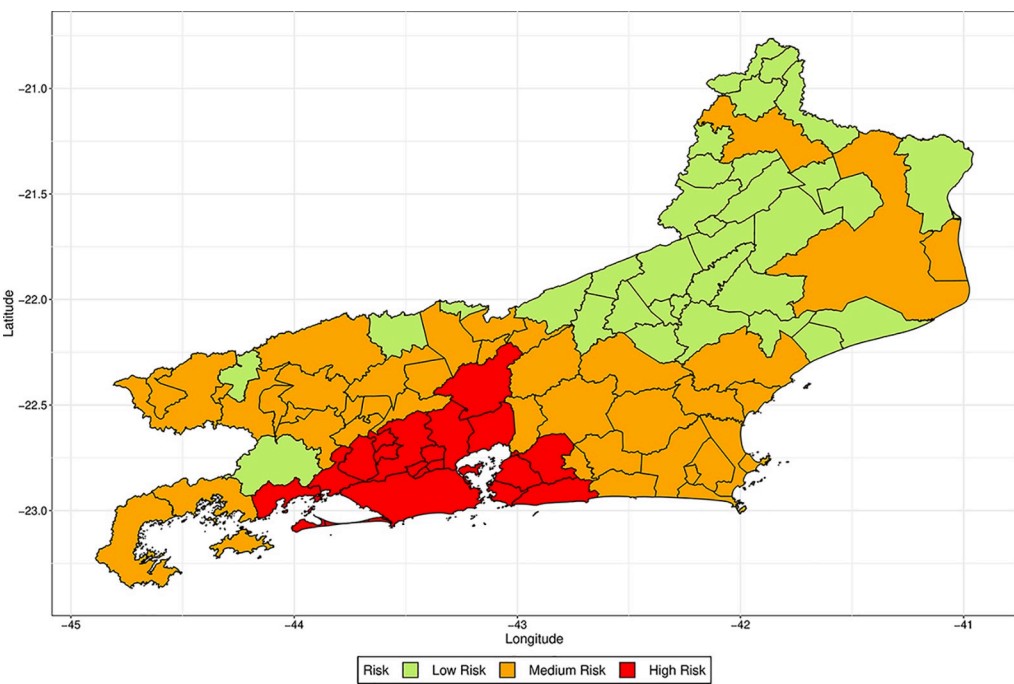

**Fig 11. Risk of each city in the state of Rio de Janeiro evaluated by k-means clustering of the ranks attributed by the simulated models with distinct values of _s_.**

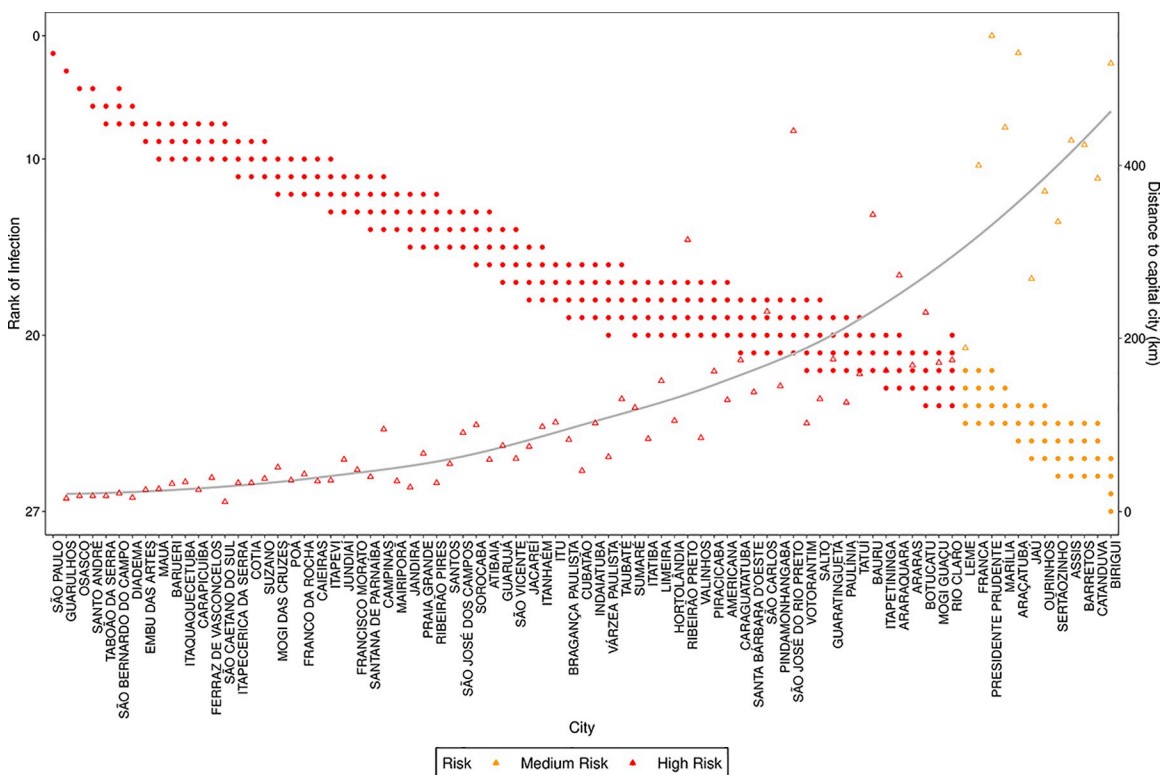

**Fig 12. Rank of infection and distance to capital city for each city with more than 100,000 inhabitants in the state of São Paulo.** The points refer to ranks estimated for different values of _s_, the triangles to the distance to the capital city and the line is a smooth approximation of the distance triangles. The colors refer to the risk evaluated by k-means clustering of the ranks attributed by the simulated models.

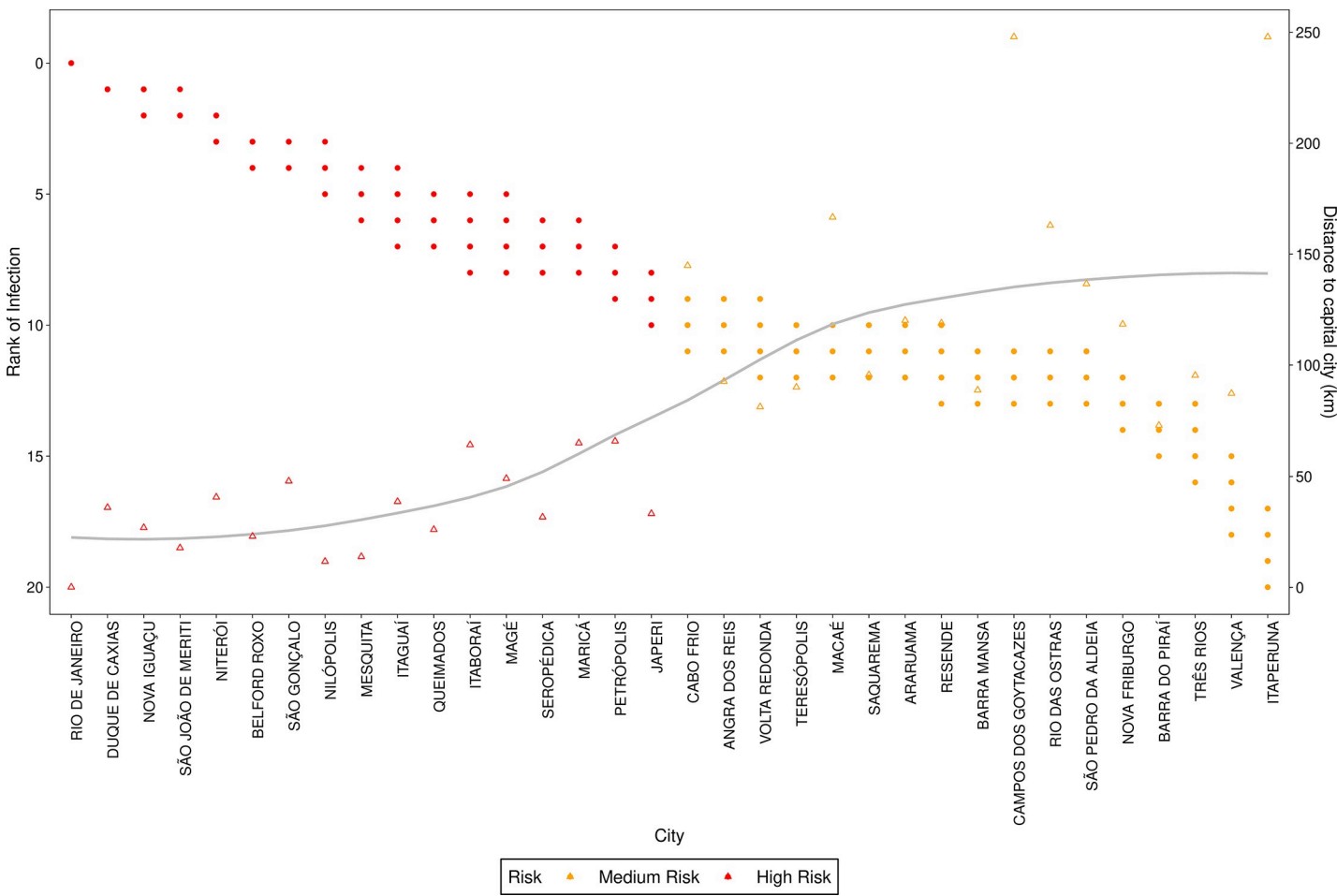

**Fig 13. Rank of infection and distance to capital city for each city with more than 75,000 inhabitants in the state of Rio de Janeiro.** The points refer to ranks estimated for different values of *s*, the triangles to the distance to the capital city and the line is a smooth approximation of the distance triangles. The colors refer to the risk evaluated by k-means clustering of the ranks attributed by the simulated models.

the locations. As expected, parameter *s* is related to the intensity of mobility, which in turn implies a greater or smaller time of infection for each city. This is an indicative that the decrease in mobility, enforced by isolation and quarantine measures, may slow the spread of the disease. Also, we proposed a risk index, based on ranks of the estimated time for an infected individual to be identified in a specific city. The risk index was shown to be robust and consistent with the spreading patterns, independent of the mobility intensity parameter *s*.

In summary, the risk model derived from the SI model seems to be robust with respect to infection rate and mobility intensity, due to the non-parametric analysis of the resulting model simulations. The risk analysis is mainly sensitive to two modelling aspects: the initial condition distribution in the network and the mobility pattern. The assumption of initial condition given by having infection only present in the capital cities is reasonable considering the observed initial appearances of the disease in Brazil. The mobility pattern assumption is endorsed by the large penetration of the mobility dataset used.

Considering the publication review process time, at the time of publication of this work the COVID19 infection counts of the simulated period were already available. We show in Fig 14 and Fig 15 an example of the adherence of the model in the risk prediction for the state of São

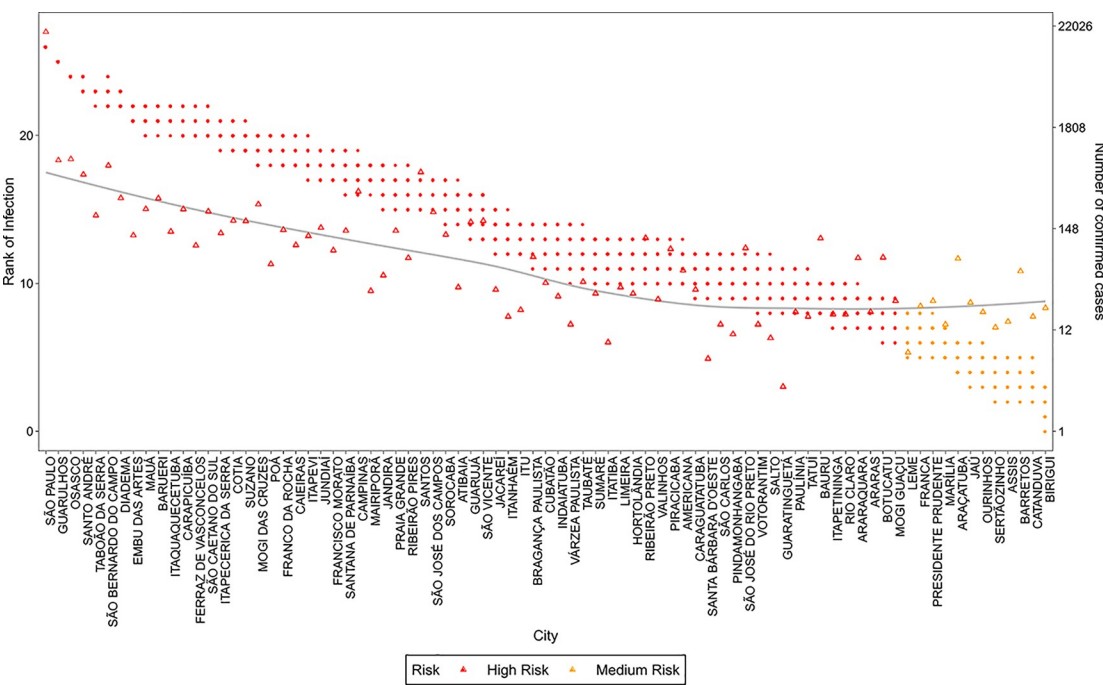

**Fig 14. Rank of infection and number of confirmed COVID19 cases for São Paulo state.** The points refer to ranks estimated for different values of *s*, the triangles refer to the official number of confirmed COVID19 cases registered on the 1st of May 2020 for each city in the state of São Paulo. The line is a smooth approximation of the confirmed cases (triangles). The colors refer to the risk evaluated by k-means clustering of the ranks attributed by the simulated models.

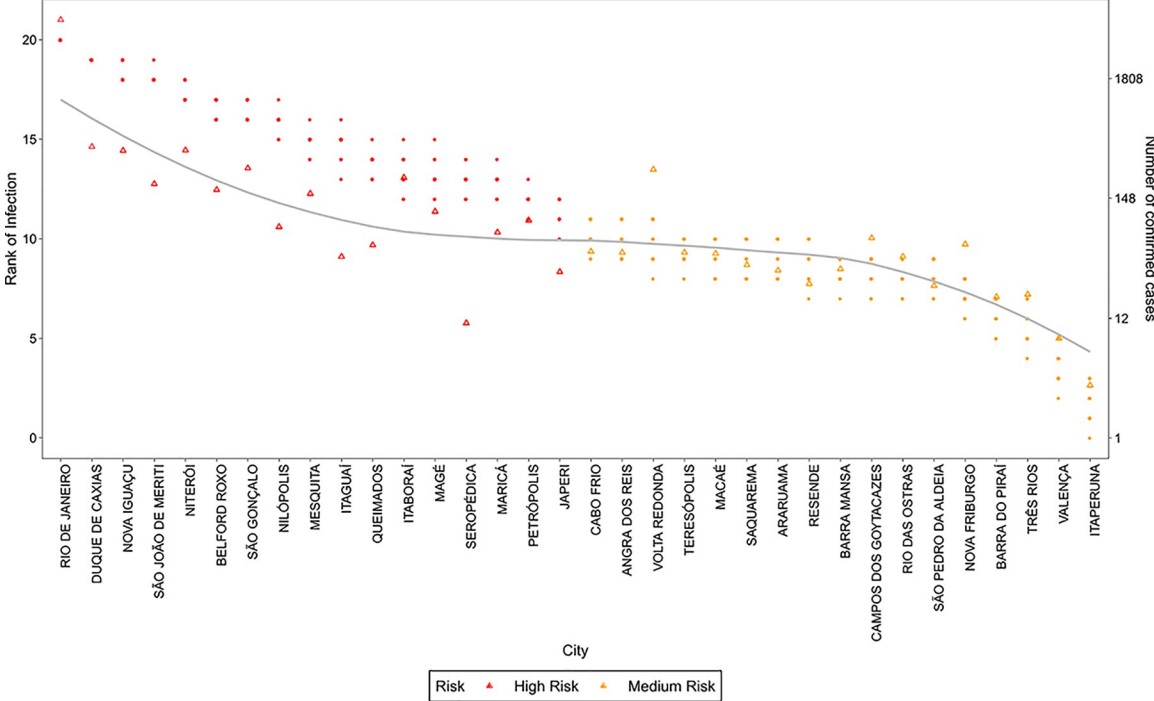

**Fig 15. Rank of infection and number of confirmed COVID19 cases for Rio de Janeiro state.** The points refer to ranks estimated for different values of *s*, the triangles refer to the official number of confirmed COVID19 cases registered on the 1st of May 2020 for each city in the state of Rio de Janeiro. The line is a smooth approximation of the confirmed cases (triangles). The colors refer to the risk evaluated by k-means clustering of the ranks attributed by the simulated models.

Paulo and Rio de Janeiro. The results show a remarkably good prediction of the spreading pattern, with only a few exceptions.

The next steps of this work are two-fold. Initially, we will extend the analysis to other states of the country and relate the infection risk to geolocated health and economic variables, to help in the planning of local financial and hospital resources allocation, and of economic loss mitigation strategies. Additionally, we will also address later phases of the disease, considering a more complex model, such as an SEIR (Susceptible—Exposed—Infectious—Recovered) coupled with mobility, allowing long term projections and better development of control measures.

## Acknowledgments

Lucas Queiroz, Rafael Gouveia and Afonso Delgado, from In Loco company, are greatly acknowledged for data acquisition and processing.

## Author Contributions

**Conceptualization:** Pedro S. Peixoto, Cláudia Peixoto, Sérgio M. Oliva.

**Data curation:** Pedro S. Peixoto.

**Formal analysis:** Pedro S. Peixoto, Diego Marcondes, Cláudia Peixoto, Sérgio M. Oliva.

**Funding acquisition:** Pedro S. Peixoto, Sérgio M. Oliva.

**Investigation:** Pedro S. Peixoto, Diego Marcondes, Cláudia Peixoto, Sérgio M. Oliva.

**Methodology:** Pedro S. Peixoto, Diego Marcondes, Cláudia Peixoto, Sérgio M. Oliva.

**Project administration:** Pedro S. Peixoto, Sérgio M. Oliva.

**Resources:** Pedro S. Peixoto, Diego Marcondes, Sérgio M. Oliva.

**Software:** Pedro S. Peixoto, Diego Marcondes.

**Supervision:** Pedro S. Peixoto, Cláudia Peixoto, Sérgio M. Oliva.

**Validation:** Pedro S. Peixoto, Diego Marcondes, Cláudia Peixoto, Sérgio M. Oliva.

**Visualization:** Pedro S. Peixoto, Diego Marcondes.

**Writing – original draft:** Pedro S. Peixoto, Diego Marcondes, Cláudia Peixoto, Sérgio M. Oliva.

**Writing – review & editing:** Pedro S. Peixoto, Diego Marcondes, Cláudia Peixoto, Sérgio M. Oliva.

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
