## [Decision Letter · Decision Letter 0]

9 Jun 2020

PONE-D-20-10452

Modeling future spread of infections via mobile geolocation data and population dynamics. An application to COVID-19 in Brazil.

PLOS ONE

Dear Dr. Peixoto,

Thank you for submitting your manuscript to PLOS ONE. After careful consideration, we feel that it has merit but does not fully meet PLOS ONE’s publication criteria as it currently stands. Therefore, we invite you to submit a revised version of the manuscript that addresses the points raised during the review process.

Your manuscript was reviewed by one expert whose comments acknowledge the merits of your paper. However,  this reviewer has also raised several points that prevent publication in the present form. Most of the issues have to do with possible limitations of your work, and I believe it can be solved by expanding your discussions. I would also like to apologize for the delay in obtaining reports on your manuscript.

We look forward to receiving your revised manuscript.

Kind regards,

Haroldo V. Ribeiro

Academic Editor

PLOS ONE

Journal Requirements:

2. Our internal editors have looked over your manuscript and determined that it is within the scope of our Cities as Complex Systems Call for Papers. This collection of papers is headed by a team of Guest Editors for PLOS ONE: Marta Gonzalez (University of California, Berkeley) and Diego Rybski (Potsdam Institute for Climate Impact Research).

The Collection will encompass a diverse and interdisciplinary set of research articles applying the principles of complex systems and networks to problems in urban science.  Additional information can be found on our announcement page: https://collections.plos.org/s/cities.

If you would like your manuscript to be considered for this collection, please let us know in your cover letter and we will ensure that your paper is treated as if you were responding to this call. If you would prefer to remove your manuscript from collection consideration, please specify this in the cover letter.

We note that one or more of the authors are employed by a commercial company: "In Loco Company,"

4. We note that Figures 3-11 in your submission contain map images which may be copyrighted. All PLOS content is published under the Creative Commons Attribution License (CC BY 4.0), which means that the manuscript, images, and Supporting Information files will be freely available online, and any third party is permitted to access, download, copy, distribute, and use these materials in any way, even commercially, with proper attribution. For these reasons, we cannot publish previously copyrighted maps or satellite images created using proprietary data, such as Google software (Google Maps, Street View, and Earth). For more information, see our copyright guidelines: http://journals.plos.org/plosone/s/licenses-and-copyright.

a) You may seek permission from the original copyright holder of Figures 3-11 to publish the content specifically under the CC BY 4.0 license. 

5. Please ensure that you refer to Figure 5 in your text as, if accepted, production will need this reference to link the reader to the figure.

Reviewers' comments:

Reviewer's Responses to Questions

**Comments to the Author**

1. Is the manuscript technically sound, and do the data support the conclusions?

Reviewer #1: Partly

2. Has the statistical analysis been performed appropriately and rigorously? 

Reviewer #1: Yes

3. Have the authors made all data underlying the findings in their manuscript fully available?

Reviewer #1: Yes

4. Is the manuscript presented in an intelligible fashion and written in standard English?

Reviewer #1: Yes

5. Review Comments to the Author

Reviewer #1: This paper investigates the spreading of SARS-CoV-2 using an SI meta-population model and mobile geolocation data. The paper is timely and addresses an important question: how to predict the number of infected people in urban areas to make better policies and prevent the spreading of COVID19? Despite its importance, the paper has several limitations. Some of these limitations are discussed by the authors but others deserve to be further discussed in the paper, as described below:

1) The positions are acquired by mobile phones with In Locos’s system. Many people don’t use a mobile phone and those who use may not have the app. How is the distribution of users in different regions? What are the limitations in terms of user-distribution across the cities?

2) The authors also aggregate the data by regions to estimate the movement between regions. Related to my previous question, because users might not be uniformly distributed because of the app usage, that would significantly affect the estimations of infected people by region depending on the scale used. What are the limitations regarding the aggregation of data by cities?

3) Models of meta-population on networks concluded that the population in each node is proportional to the number of connections of that node, i.e, population ~ k/<k>, where k is the number of connections between two regions. In some cases, there is no complete information about mobility between nodes (regions/cities), and researchers use self-loops to keep the population at a stationary level similar to the real one, otherwise, the population could increase or decrease to a stationary level different from the real value. See for instance:

Colizza, Vittoria, Romualdo Pastor-Satorras, and Alessandro Vespignani. "Reaction-diffusion processes and metapopulation models in heterogeneous networks." Nature Physics 3.4 (2007): 276-282.

Pastor-Satorras, Romualdo, et al. "Epidemic processes in complex networks." Reviews of modern physics 87.3 (2015): 925.

Vespignani, Alessandro. "Modelling dynamical processes in complex socio-technical systems." Nature Physics 8.1 (2012): 32-39.

How does your model solve this problem? It is not clear whether the correction you propose is related to these questions. Can you clarify the correction you did to the over/underestimation of movement between the two regions?

4) Another point that is not clear is the role of the time window of 24 hours. Some people could go to work for a few days in a city and return after more than 24 hours. You have that information but decided no to use, why? What are the implications of limiting the time window in 24 hours?

5) The SEIR model would be more appropriated in this analysis than the SI model. However, for short time scales, that might work, as already pointed out by the authors. What are the other limitations of the predictions due to the number of initial cases (seed of infection), the estimation of the reproductive number R0, and the mobility data itself?

6) Finally, the authors could elucidate in the discussion of what is the main novelty compared to previous finds using the SIR model and mobility data. Although the results are very interesting, I am not convinced that there is enough novelty in the paper. Can the authors comment on the discussion what are the points of the results that are in agreement with previous finds of different countries and what are the limitations of the study in making accurate predictions of the number of infected people?

After revising the points above and discussing the limitations of the paper, I have no reason to not recommend this paper for publication in PLOS ONE.

Minor point:

- The population of Rio de Janeiro State is approximately 16 millions, not 6 millions.

  </k>

6. PLOS authors have the option to publish the peer review history of their article (what does this mean?). If published, this will include your full peer review and any attached files.

Reviewer #1: No

---

## [Author Response · Author response to Decision Letter 0]

16 Jun 2020

Responses to Reviewer #1 are given bellow (indicated by > )

Reviewer #1: This paper investigates the spreading of SARS-CoV-2 using an SI meta-population model and mobile geolocation data. The paper is timely and addresses an important question: how to predict the number of infected people in urban areas to make better policies and prevent the spreading of COVID19? Despite its importance, the paper has several limitations. Some of these limitations are discussed by the authors but others deserve to be further discussed in the paper, as described below:

1) The positions are acquired by mobile phones with In Locos’s system. Many people don’t use a mobile phone and those who use may not have the app. How is the distribution of users in different regions? What are the limitations in terms of user-distribution across the cities?

> The reviewer is right to point out this limitation of the mobile data and it is indeed a matter of concern with respect to the given data. We will better explain the dataset here and we have addressed the point in the new version of the paper too, highlighting that even thought there is a limitation, the dataset still captures very well the mobility patterns between Brazilian cities.

The company holds a dataset the approximately follows 60 million mobile phones in Brazil. To date, Brazil has approximately 225 million mobile phones, according to the governmental Brazilian telecommunications agency (ANATEL –https://www.anatel.gov.br/paineis/acessos/telefonia-movel). While the company holds geolocation information of more than one fourth of the mobile phones in Brazil, this is indeed a dataset with specific population target group. The company focuses on adult apps, mainly e-commerce (the app names are held confidential, due to contract and privacy reasons). The users are estimated to be mainly on the ages between 18 and 60 years. As a reference, the Brazilian Institute of Geography and Statistics (IBGE), main provider of data and information about Brazil, estimates that the Brazilian population of this age group is approximately 125 million people. 

Therefore, while the dataset is biased towards young adults and smartphone users, its penetration is so vast that it is able to capture very well mobility patterns compared, for example, from official road transport data (IBGE), as well as inner city pendular movements, but provides richer details, as it is can be accurate to a few meters. To the best of our knowledge, this is by far the largest mobility data available for research of the population in Brazil in terms of penetration into the interior of the country (see Fig 3 of the manuscript for example) and social stratification.

To address the reviewer’s concern, we have included a disclaimer in the manuscript explaining the limitation of the dataset and further information on the matter. See Section “Methods”, subsection “Dataset“.

2) The authors also aggregate the data by regions to estimate the movement between regions. Related to my previous question, because users might not be uniformly distributed because of the app usage, that would significantly affect the estimations of infected people by region depending on the scale used. What are the limitations regarding the aggregation of data by cities?

> Along with the response to the previous question, indicating that even though there is indeed a bias in the dataset sampling with respect to the actual population, we again point out that the penetration of the company dataset is vast enough to capture the main mobility patterns of interest for the risk assessment discussed in this manuscript. In this sense, the city aggregation should not have a big influence in the regional mobility estimates. It is important to note that we are only using relative mobility patterns in the model, but not the actual flux intensity. Due to the size of the dataset, the mobility patterns are robust with respect to the small perturbation of the local fluxes.

Also, the city aggregation seems to provide a good trade-off between data availability and the population density distribution within the states, as higher resolution divisions leads to regions with relatively small samples of mobile users from the data base, and lower resolutions lead to loss of important mobility patterns responsible for the disease dissemination.

Additional information on this point was included in the new version of the manuscript in Section “Methods”, subsection “Movement dynamics”.

3) Models of meta-population on networks concluded that the population in each node is proportional to the number of connections of that node, i.e, population ~ k/, where k is the number of connections between two regions. In some cases, there is no complete information about mobility between nodes (regions/cities), and researchers use self-loops to keep the population at a stationary level similar to the real one, otherwise, the population could increase or decrease to a stationary level different from the real value. See for instance:

Colizza, Vittoria, Romualdo Pastor-Satorras, and Alessandro Vespignani. "Reaction-diffusion processes and metapopulation models in heterogeneous networks." Nature Physics 3.4 (2007): 276-282.

Pastor-Satorras, Romualdo, et al. "Epidemic processes in complex networks." Reviews of modern physics 87.3 (2015): 925.

Vespignani, Alessandro. "Modelling dynamical processes in complex socio-technical systems." Nature Physics 8.1 (2012): 32-39.

How does your model solve this problem? It is not clear whether the correction you propose is related to these questions. Can you clarify the correction you did to the over/underestimation of movement between the two regions?

> Thank you for pointing this out. We are aware of these references and this issue. We will highlight here the main points related to this concern and we have also added further description in the new version of the manuscript discussing the matter.

> First, and interestingly, our mobility data shows a close to symmetric structure on a time average. So, even though at a certain day it may show large flux in one direction, on another day this is compensated by flux in the opposite direction. In this sense, for the time period considered in this paper, no important population increase/decrease was observed. Additionally, in our model, the network self-loops (matrix diagonal) data are only used as a normalization factor that influences the intensity of the fluxes, but not the flux pattern on the network. To account for this possible overestimation of the flux intensity on the network we have added an additional parameter in the non-local spatial spreading of the model.

> We understand that the preservation of the stationary population level is a main issue in analysis for longer periods of time, which is beyond the scope of this manuscript, but may be addressed in the follow up work. 

> The new version of the manuscript now points out in the introduction some related research on epidemic models on complex networks, and the connections and differences with respect to the current work. We point out that this work is data driven and is heuristic in nature, extracting the mobility pattern between cities directly from data. To provide robustness to the risk model, we included an additional parameter to account for possible city wise flux intensity errors relative to local dissemination of the disease, mainly ensuring preservation of the mobility pattern and only compensating for flux intensity changes. This ensures that the on a short time scale the model is robust and qualitatively an adequate representation of the disease spread within the cities of the states considered in the study.

>Additional information about this issue was included in the new version of the manuscript. See Introduction and Methods/Movement Dynamics

4) Another point that is not clear is the role of the time window of 24 hours. Some people could go to work for a few days in a city and return after more than 24 hours. You have that information but decided no to use, why? What are the implications of limiting the time window in 24 hours?

> Thank you for point this out, it was indeed unclear in the original manuscript. In the case mentioned, of someone going to work for a few days in a city returning only after a period longer than 24h, there would be 2 trips registered in the database: one relative to the outgoing on a certain day, and then the other relative to the return, on the other day, separately. So, this case is considered in the analyzed dataset. 

> The limitations of the 24h time window are as follows. A user that was steady for 24h, without using the mobile phone, only to use the mobile on the next day in a different place, would be considered a steady user (no movement within the day under analysis). A user that performed an exceptionally long (over 24h) journey, without stops and with an offline mobile, would be excluded from the analysis. Due to the very frequent (order of seconds) data collection mechanism of the SDK kit and the collection possible while in background mode (even while not using a specific app with the SDK), both events are rare. Therefore, this filtering was employed basically to ensure a consistent dataset with well defined movement time window frame. 

> To address this point, we have added a sentence on the new version of the manuscript better explaining the time window implications, which are minimal. Refer to “Methods” section, subsection “Dataset”.

5) The SEIR model would be more appropriated in this analysis than the SI model. However, for short time scales, that might work, as already pointed out by the authors. What are the other limitations of the predictions due to the number of initial cases (seed of infection), the estimation of the reproductive number R0, and the mobility data itself?

> The decision for using a simpler model, SI, instead of the SEIR model, was to ensure that the initial stages of the exponential spreading of the disease was well captured with minimal influence from the uncertainty of other parameters present in multiparametric models such as SEIR. In the proposed model, we have only 2 parameters: one directly connected to the reproductive number (the infection rate) and the other related to the strength of the mobility flow, allowing a clear analysis of local and regional disease growth in its initial stage. The SEIR model carries additional complexity that is unnecessary in the initial stages of the propagation, including more parameters subject to imprecisions. 

> Additionally, the risk model derived from the SI model seems to be robust with respect to infection rate and mobility intensity, due to the non-parametric analysis of the resulting model simulations. The risk analysis is mainly sensitive to two modelling aspects: the initial condition distribution in the network and the mobility pattern. The assumption of initial condition given by having infection only present in the capital cities is reasonable considering the observed initial appearances of the disease in Brazil. The mobility pattern assumption is endorsed by the large penetration of the mobility dataset used.

> We are currently investigating a multi-compartment metapopulation model (SEIR with added compartments) for later stages of the COVID19 dissemination. This model has increased complexity when coupled with spatial mobility, due to the heterogenic nature of the regions, which are under different stages of the epidemic curve. Several parameters need to be inferred to ensure an accurate temporal-spatial metapopulation model of this nature; therefore, this is beyond the scope of this manuscript and will be presented in a companion paper.

6) Finally, the authors could elucidate in the discussion of what is the main novelty compared to previous finds using the SIR model and mobility data. Although the results are very interesting, I am not convinced that there is enough novelty in the paper. Can the authors comment on the discussion what are the points of the results that are in agreement with previous finds of different countries and what are the limitations of the study in making accurate predictions of the number of infected people?

> The main novelty of the manuscript is not necessarily related to methodological modeling aspects, but rather related to its practical importance and urgency of the public health matter. Also, this mobility dataset is a huge source of rich information on public mobility, providing means of evaluating the efficacy of epidemiological models on complex networks on a realistic scenario. This paper shows how a simple SI model on a complex network can adequately represent a complex realistic scenario, mainly possible only due to the richness of the network data at hand.

> Additional information about connections with findings for different countries are better described in the Introduction of the new version of the manuscript. While the results agree with existing literature, we point out that, to the best of our knowledge, existing studies on spatial-temporal network metapopulation models, especially for COVID19, use static regional mobility data sets, provided via government census, for example, or global wide international air traffic data. The usage of such a detailed and vast mobile mobility dataset, as the one under investigation in this paper, seems to have been rarely explored in this modeling framework. This sort of dataset exists on a global scale, for example considering the geolocation data collected by Google and Apple, and other mobile service providers, such as In Loco company. However, the available data of such companies are usually limited to social distancing information and aggregated urban mobility human behavior patterns, providing measures of quarantine efficacy and control. Mobile regional travel information is very rarely available for research purposes, which helps to explain the lack of existing research on a similar basis as the one provided in this work.

> The version of the manuscript contains clarifications on this matter both in the Introduction and Discussion sections.

After revising the points above and discussing the limitations of the paper, I have no reason to not recommend this paper for publication in PLOS ONE.

> We hope to have addressed all the concerns raised by the reviewer in the new version of the manuscript.

Minor point:

- The population of Rio de Janeiro State is approximately 16 millions, not 6 millions.

> Thank you for pointing this.

---

## [Decision Letter · Decision Letter 1]

23 Jun 2020

Modeling future spread of infections via mobile geolocation data and population dynamics. An application to COVID-19 in Brazil.

PONE-D-20-10452R1

Dear Dr. Peixoto,

We’re pleased to inform you that your manuscript has been judged scientifically suitable for publication and will be formally accepted for publication once it meets all outstanding technical requirements.

Kind regards,

Haroldo V. Ribeiro

Academic Editor

PLOS ONE

Reviewers' comments:

Reviewer's Responses to Questions

**Comments to the Author**

1. If the authors have adequately addressed your comments raised in a previous round of review and you feel that this manuscript is now acceptable for publication, you may indicate that here to bypass the “Comments to the Author” section, enter your conflict of interest statement in the “Confidential to Editor” section, and submit your "Accept" recommendation.

Reviewer #1: All comments have been addressed

2. Is the manuscript technically sound, and do the data support the conclusions?

Reviewer #1: Yes

3. Has the statistical analysis been performed appropriately and rigorously? 

Reviewer #1: Yes

4. Have the authors made all data underlying the findings in their manuscript fully available?

Reviewer #1: Yes

5. Is the manuscript presented in an intelligible fashion and written in standard English?

Reviewer #1: Yes

6. Review Comments to the Author

Reviewer #1: In general, I am happy with the answers provided by the authors and I have no other reason to not recommend this paper for publication in PLOS ONE.

7. PLOS authors have the option to publish the peer review history of their article (what does this mean?). If published, this will include your full peer review and any attached files.

Reviewer #1: No

---

## [Editor Report · Acceptance letter]

8 Jul 2020

PONE-D-20-10452R1 

Modeling future spread of infections via mobile geolocation data and population dynamics. An application to COVID-19 in Brazil. 

Dear Dr. Peixoto:

I'm pleased to inform you that your manuscript has been deemed suitable for publication in PLOS ONE. Congratulations! Your manuscript is now with our production department. 

Kind regards, 

on behalf of

Dr. Haroldo V. Ribeiro 

Academic Editor

PLOS ONE